# Continuous Thought Machines

Luke Darlow[1]  Ciaran Regan[1,2]  Sebastian Risi[1,3]  Jeffrey Seely[1]  Llion Jones[1]

[1]Sakana AI, Tokyo, Japan
[2]University of Tsukuba, Japan
[3]IT University of Copenhagen, Denmark
{luke, ciaran, sebastianrisi, jeffrey, llion}@sakana.ai

## Abstract

Biological brains demonstrate complex neural activity, where neural dynamics are critical to how brains process information. Most artificial neural networks ignore the complexity of individual neurons . We challenge that paradigm. By incorporating neuron-level processing and synchronization, we reintroduce neural timing as a foundational element. We present the Continuous Thought Machine (CTM), a model designed to leverage neural dynamics as its core representation. The CTM has two innovations: (1) **neuron-level temporal processing**, where each neuron uses unique weight parameters to process incoming histories; and (2) **neural synchronization as a latent representation**. The CTM aims to strike a balance between neuron abstractions and biological realism. It operates at a level of abstraction that effectively **captures essential temporal dynamics while remaining computationally tractable**. We demonstrate the CTM's performance and versatility across a range of tasks, including solving 2D mazes, ImageNet-1K classification, parity computation, and more. Beyond displaying rich internal representations and offering a natural avenue for interpretation owing to its internal process, the CTM is able to perform tasks that require complex sequential reasoning. The CTM can also leverage adaptive compute, where it can stop earlier for simpler tasks, or keep computing when faced with more challenging instances. The goal of this work is to share the CTM and its associated innovations, rather than pushing for new state-of-the-art results. To that end, we believe the CTM represents a significant step toward developing more biologically plausible and powerful artificial intelligence systems. We provide an accompanying **interactive online demonstration** and an **extended technical report**.

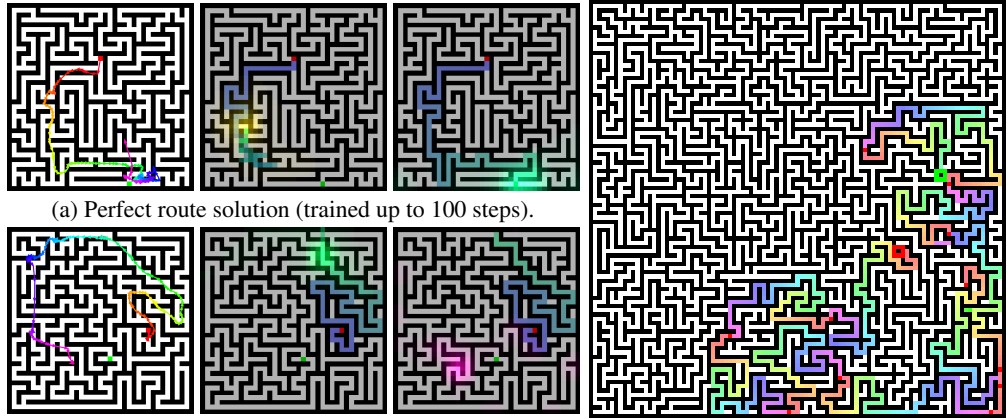

(a) Perfect route solution (trained up to 100 steps).

(b) (**Emergent**) further than the training route.

(c) Generalizing to a larger maze.

Figure 1: Solving 100 steps down $39 \times 39$ mazes. (a, b) Observing using attention (**no positional encoding**, weights overlaid), imagining a route (arrows) from red to green pixels, (b) attending beyond 100 steps, and (c) generalizing to $99 \times 99$ via sequential re-applications of the same model.

39th Conference on Neural Information Processing Systems (NeurIPS 2025).

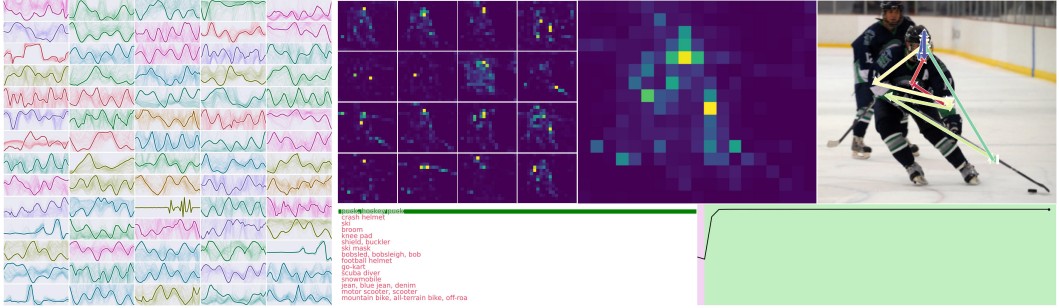

(a) Each random-colored subplot is a **single neuron's activity**.

(b) The CTM looks around to build up its prediction, effectively tracing an intuitive path by **synchronizing its neurons to attend dynamically**.

Figure 2: ImageNet-1K demonstration. (a) Complex neural dynamics whose **synchronization are the representation** with which the CTM observes and predict. (b) CTM's **attention process**, showing all 16 attention heads (left) and average thereof (middle). Arrows trace the average weighting over internal ticks, exemplifying a complex path that emerges without any training signal. We discuss more interesting emergent properties of the CTM in Appendix I. Video demonstrations are **here**.

# 1   Introduction

Biological brains exhibit complex time-dependent neural dynamics, but artificial neural networks (NNs) intentionally abstract away the precise timing and interplay of neuron interactions to facilitate large-scale deep learning [1, 2, 3]. While enabling significant advancements over the years, these simplifications deviate from fundamental biological neural computation principles. Emulating the temporal aspects of neural dynamics present in brains remains challenging. Consequently, modern NNs prioritize simplicity and computational efficiency over strict emulation. This abstraction, though task-performant, contributes to a gap between flexible human cognition and current AI capabilities, suggesting missing fundamental components, potentially related to temporal processing [4, 5, 6].

Despite its outstanding performance, modern AI lacks the flexibility, efficiency, fluidity, generalization capabilities, and common sense of human intelligence, which operates in an open world where learning and adaptation are tied to the arrow of time [5, 7, 6, 8]. We argue that incorporating time as part of neural computation is crucial for advancing AI [9, 10]. We introduce the *Continuous Thought Machine* (CTM), a model explicitly incorporating neural dynamics over time. Our contributions are:

1. The CTM architecture  using an **internal dimension**  for modeling the temporal evolution of neural activity, **neuron-level models** (NLMs) as a more biologically plausible mid-level abstraction of neurons that unfold neural dynamics , and the use of **neural synchronization** directly as the representation (implemented via temporal correlations between neuron-level activity; Section 3.4)  for observation and prediction, making neural dynamics the core operating principle.

2. An exposition of the capabilities unlocked by the CTM, including strong performance on sequential  reasoning tasks (Figure 1) , native adaptive compute time, natural and interpretable behaviors such as 'looking around' images before predicting (Figure 2) , and learning algorithmic solutions, opening up opportunities to the AI community for new research.

The CTM learns to use neural synchronization as its latent representation, distinguishing it from existing work that explores synchrony as emergent properties for post-hoc use [11, 12]. This representation is distinct from the common static 'snapshot' representations used in most modern NNs as it directly encodes the temporal interplay of neural dynamics.

**Recurrence and Reasoning.**   Recurrence is a strong contender for extending model complexity beyond current scaling limitations [13, 14, 15]. We posit that recurrence, while essential, is merely one piece of the puzzle. The temporal dynamics unlocked by recurrence are equally crucial. We demonstrate in this paper that neural dynamics can be leveraged to build a new kind of neural network with surprising capabilities. We show how the CTM navigates complex 2D mazes by forming internal maps without positional encodings (Section 4), learns to 'look around' (without any signal to do so) when classifying images and exhibits native adaptive computation time as a side-effect

(Section 5), and utilizes its dynamic representations for tasks requiring memory and sequential reasoning (Section 6). These capabilities emerge from the same core architecture applied to different tasks, showcasing its versatility and trainability. We believe that the CTM represents a step towards bridging the gap between powerful modern AI and biological plausibility.

The remainder of this paper details related work (Section 2), describes the CTM (Section 3), evaluates core capabilities on 2D mazes, ImageNet-1K classification, and parity computation (Sections 4 to 6), summarizes further experiments and applications (Section 7), and discusses findings (Section 8).

## 2 Related Work

The CTM uses neural timing and synchronization as core computational principles. This positions it relative to, yet distinct from, several lines of research.

**Adaptive Computation.** Many approaches achieve adaptive computation via explicit mechanisms. Early-exit networks [16] use intermediate classifiers for early termination. PonderNet [17] and Adaptive Computation Time (ACT) [18] introduce learnable halting modules governing recurrent steps. More recent methods like AdaTape [19] dynamically extend input sequences, while Sparse Universal Transformers (SUT) [20] combine recurrent weight sharing with dynamic halting and Mixture-of-Experts. In contrast, the CTM's adaptive processing (varying internal ticks per input based on certainty and loss dynamics; Section 3.5) emerges naturally from its core architecture, driven by the unfolding of its internal neural dynamics without dedicated halting components.

**Iterative and Recurrent Reasoning.** The CTM's internal ticks facilitate iterative refinement, akin to models promoting internal computational steps. For instance, Quiet-STaR [21] uses hidden rationale generation in language models, and Recurrent Independent Mechanisms (RIMs) [22] employ modular, asynchronous sub-networks for multi-step reasoning. While Recurrent Models of Visual Attention (RAM) [23] also leveraged recurrence for sequential processing of visual glimpses, the CTM's novelty lies in generating internal neural dynamics from neuron-level histories across a decoupled time dimension and then utilizing the **explicit temporal patterns of neural synchronization** as its primary representation. This contrasts with RAM's focus on perceptual decision-making from external glimpses or models relying solely on a final recurrent state.

**Biologically Inspired Neural Dynamics.** There is growing interest in more biologically plausible neural computation [24]. Examples include Liquid Time-Constant Networks (LTCNs) [25] with neurons governed by time-varying differential equations, and various Spiking Neural Network (SNN) paradigms that inherently use discrete, timed events, with recent work also exploring synchronization mechanisms [26, 27]. The CTM draws inspiration from temporal coding and neural synchrony, but uses: (1) neuron-level models (NLMs) to process a history of continuous-valued pre-activations to generate complex dynamics, and (2) *neural synchronization* as the primary latent representation for attention and output. While inspired by principles like spike-timing and synchrony, CTM abstracts these—focusing on local temporal integration and population-level synchronization—into a tractable, differentiable framework suitable for gradient-based deep learning, rather than replicating detailed biophysics. This situates the CTM alongside, yet distinct from, extensive work on models such as Liquid State Machines [28], and diverse SNNs that exploit precise spike timing for computation or employ specialized learning rules [29, 30, 31, 32, 33]. These latter models often emphasize event-driven dynamics, explore non-differentiable computation, or focus on online learning. The CTM offers a complementary direction, retaining inspiration from biological timing while ensuring compatibility with established deep learning training paradigms.

**Synchronization.** Reichert & Serre [11] proposed a model where synchronization emerges from interactions among complex-valued neurons, serving as a gating mechanism that modulates information flow and enables post-hoc grouping of neurons for tasks like object segmentation. Unlike CTM, however, their model does not use synchrony as a learned latent representation during computation. Other approaches in complex-valued neural networks [12] employ synchronization from a control-theoretic perspective, aiming to stabilize or coordinate networks via externally enforced synchrony. In contrast, CTM integrates synchronization intrinsically, optimizing neural phase relationships during training to encode task-relevant representations. This positions CTM as a computationally grounded model of synchrony, fundamentally distinct from prior works that treat synchrony as a control objective.

# 3 Method

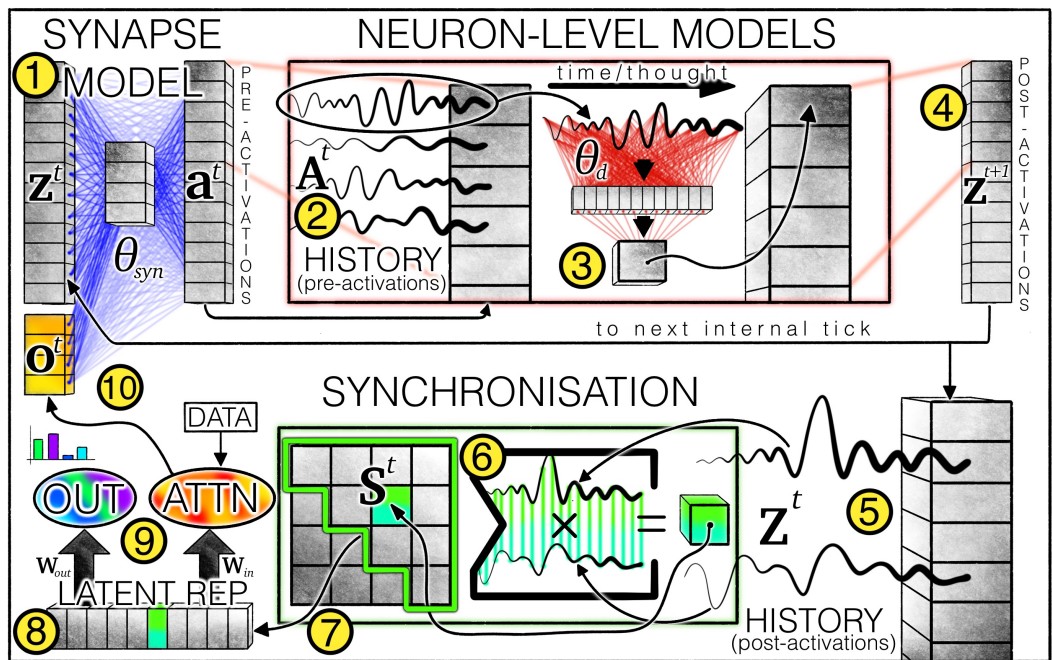

Figure 3: CTM architecture overview. Key components include: ① Synapse model generating pre-activations from prior post-activations $\mathbf{z}^t$ and attention output $\mathbf{o}^t$. ② History of pre-activations $\mathbf{A}^t$. ③ Neuron-level models (NLMs) processing $\mathbf{A}_d^t$ to produce ④ post-activations $\mathbf{z}_d^{t+1}$. ⑤ History of post-activations $\mathbf{Z}^t$. ⑥ Neural synchronization matrix $\mathbf{S}^t$ computed from $\mathbf{Z}^t$. ⑦ Selected neuron pairs from $\mathbf{S}^t$ form ⑧ latent representations used for ⑨ outputs $\mathbf{y}^t$ and attention queries $\mathbf{q}^t$. ⑩ Attention output $\mathbf{o}^t$ is concatenated with $\mathbf{z}^{t+1}$ for the next internal tick. Owing to the inherent difficulty in visualizing a dynamic, time-based architecture, we include the supplementary video '*arch.mp4*' (hosted **here** too) that visualizes functional data flow.

The Continuous Thought Machine (CTM) is a neural network architecture that explicitly incorporates **neural dynamics** as a core component. Figure 3 ① → ⑩ and pseudocode in Listing 1 illustrate the CTM's flow. The CTM differs from other recurrent architectures [34, 35, 36, 18, 37] in two ways: (1) it applies **neuron-level models** (NLMs), each with private weights, to histories of pre-activations to produce complex neuron-level activity (Section 5); and (2) it uses **neural synchronization directly** as the latent representation for modulating data and producing outputs (Section 3.4).

## 3.1 Continuous Thought: The Internal Sequence Dimension

The CTM uses an internal dimension $t \in \{1, \ldots, T\}$, decoupled from data dimensions. This timeline of internal ticks [34, 35, 36, 37] enables iterative refinement of representations, even for static data. Unlike conventional sequential models that process data-inherent sequences, the CTM along a self-generated timeline of 'thought steps' that unfolds neural dynamics for downstream use.

## 3.2 Recurrent Weights: Synapses

A ① synapse model, $f_{\theta_{\text{syn}}}$, interconnects neurons in a shared $D$-dimensional latent space, $\mathbf{z}^t \in \mathbb{R}^D$. We found a U-NET-esque [38] MLP (details in Appendix C.1) performs best, suggesting benefit from deeper and more flexible synaptic computation. It produces **pre-activations**, $\mathbf{a}^t$:

$$\mathbf{a}^t = f_{\theta_{\text{syn}}}(\text{concat}(\mathbf{z}^t, \mathbf{o}^t)) \in \mathbb{R}^D, \tag{1}$$

where $\mathbf{o}^t$ is attention output (Section 3.4). The $M$ most recent pre-activations form a ② history $\mathbf{A}^t$:

$$\mathbf{A}^t = \begin{bmatrix} \mathbf{a}^{t-M+1} & \mathbf{a}^{t-M+2} & \cdots & \mathbf{a}^t \end{bmatrix} \in \mathbb{R}^{D \times M}. \tag{2}$$

Initial pre-activation history and $\mathbf{z}^{t=1}$ are learnable parameters. We found that setting $M \approx 10 - 100$ was effective during our initial exploration.

### 3.3 Privately-Parameterized Neuron-Level Models (NLMs)

Each neuron $d \in \{1, \ldots, D\}$ has a ③ privately parameterized NLM, $g_{\theta_d}$ (depth 1 MLP of width $d_{\text{hidden}}$), processing its $M$-dimensional pre-activation history $\mathbf{A}_d^t$ to produce ④ post-activations:

$$\mathbf{z}_d^{t+1} = g_{\theta_d}(\mathbf{A}_d^t). \tag{3}$$

The full set of post-activations $\mathbf{z}^{t+1}$ is ⑩ concatenated with attention output, $\mathbf{o}^t$, and fed into the synapse model $f_{\theta_{\text{syn}}}$ for the next internal tick, $t + 1$. See Listing 2 for pseudo-code.

### 3.4 Neural Synchronization: Modulating Data and Outputs

Synchronization is inspired by biological brains [39]. The CTM modulates data via the **synchronization of neural activity**[1]. We first collect post-activations into ⑤ a (non-fixed length) history:

$$\mathbf{Z}^t = \begin{bmatrix} \mathbf{z}^1 & \mathbf{z}^2 & \cdots & \mathbf{z}^t \end{bmatrix} \in \mathbb{R}^{D \times t}. \tag{4}$$

We define neural synchronization is defined as the ⑥ inner product of the histories of each neuron:

$$\mathbf{S}^t = \mathbf{Z}^t \cdot (\mathbf{Z}^t)^{\mathsf{T}} \in \mathbb{R}^{D \times D}. \tag{5}$$

#### 3.4.1 Neuron Pairing: A Sub-sampling Approach

Since $\mathbf{S}^t$ scales with $O(D^2)$ it can grow very large. We sample $(i, j)$ neurons at the start of training by randomly selecting $D_{\text{out}}$ and $D_{\text{action}}$ pairs for two **synchronization representations**, $\mathbf{S}_{\text{out}}^t \in \mathbb{R}^{D_{\text{out}}}$ and $\mathbf{S}_{\text{action}}^t \in \mathbb{R}^{D_{\text{action}}}$. These are projected by $\mathbf{W}_{\text{out}}$ and $\mathbf{W}_{\text{in}}$ for outputs $\mathbf{y}^t$ and attention queries $\mathbf{q}^t$:

$$\mathbf{y}^t = \mathbf{W}_{\text{out}} \cdot \mathbf{S}_{\text{out}}^t, \tag{6}$$

$$\mathbf{q}^t = \mathbf{W}_{\text{in}} \cdot \mathbf{S}_{\text{action}}^t. \tag{7}$$

We use standard cross attention [40] for $\mathbf{o}^t$:

$$\mathbf{o}^t = \text{Attention}(Q = \mathbf{q}^t, KV = \text{FeatureExtractor}(\text{data})), \tag{8}$$

where a FeatureExtractor (e.g., ResNet [41]) provides keys/values. $\mathbf{o}^t \in \mathbb{R}^{d_{\text{input}}}$ is then concatenated with $\mathbf{z}^{t+1}$. This process, including learnable temporal scaling, is shown in Listing 3.

**Scaling Temporal Dependency.** To modulate the influence of past activity on $\mathbf{S}^t$, we introduce learnable exponential decay factors $r_{ij} \geq 0$ for each neuron pair $ij$. The rescaling vector over $t$ is:

$$\mathbf{R}_{ij}^t = \begin{bmatrix} \exp(-r_{ij}(t-1)) & \exp(-r_{ij}(t-2)) & \cdots & \exp(0) \end{bmatrix}^{\mathsf{T}} \in \mathbb{R}^t. \tag{9}$$

The rescaled synchronization is (see Appendix H for efficient recursive computation):

$$\mathbf{S}_{ij}^t = \frac{(\mathbf{Z}_i^t)^{\mathsf{T}} \cdot \text{diag}(\mathbf{R}_{ij}^t) \cdot (\mathbf{Z}_j^t)}{\sqrt{\sum_{\tau=1}^t [\mathbf{R}_{ij}^t]_\tau}}. \tag{10}$$

Higher $r_{ij}$ bias towards recent ticks ($r_{ij} = 0$ means no decay). Learnable decay rates $r_{ij}$ allow the CTM to modulate synchronization across multiple time scales[2]. Details on neuron-pair sub-sampling strategies, including recovering snapshot dependencies, are in Appendix C.2.

### 3.5 Loss Function: Optimizing Across Internal Ticks

The CTM produces outputs $\mathbf{y}^t \in \mathbb{R}^C$ (e.g., class probabilities) at each internal tick $t$. We compute a loss $\mathcal{L}^t = \text{CrossEntropy}(\mathbf{y}^t, y_{true})$ and certainty $\mathcal{C}^t$ (1 - normalized entropy) per tick. For each forward pass we select to ticks:

1. the point of minimum loss: $t_1 = \text{argmin}(\mathcal{L})$, to optimize the 'best' prediction; and

---

[1]We found that 'snapshot' representations were too constraining: projecting from $\mathbf{z}^t$ strongly ties it to the downstream task and thereby limits the types of dynamics it can produce, whereas synchronization decouples it.

[2]For full disclosure, we found that the CTM barely leveraged this for ImageNet (Section 5) but more so for 2D mazes (Section 4), suggesting task-dependent temporal sensitivities

2. the point of maximum certainty: $t_2 = \mathrm{argmax}(\mathcal{C})$, to ensure certainty aligns with correctness.

The final loss for optimizing $\theta_{\mathrm{syn}}$ and $\theta_{d=1\ldots D}$ is:

$$L = \frac{\mathcal{L}^{t_1} + \mathcal{L}^{t_2}}{2}. \tag{11}$$

Since $t_1$ and $t_2$ are dynamically defined per data point, the CTM can attribute variable compute (internal ticks) to different data points as needed **without explicit restrictions** on which tick should be used in the loss function. This effectively implements **native adaptive computation** [18] as opposed to a post-hoc addition. We give pseudo-code in Listing 4.

## Experimental Evaluation

The following sections present a focused evaluation of the CTM on tasks that highlight its core principles: **neuron-level temporal processing** and **neural synchronization as a direct latent representation**. We aim to demonstrate how neural dynamics enables the CTM to implement complex reasoning or adaptive processing, while yielding interpretable strategies. We prioritize depth in three key experiments: 2D maze navigation, ImageNet-1K classification, and parity computation. We also summarize and highlight additional experiments demonstrating the CTM's broader capabilities.

## 4   2D Mazes: Complex Sequential Reasoning and Internal World Models

In this section we analyze the CTM's capacity for sequential reasoning, planning, and spatial understanding using a challenging phrasing of the 2D maze navigation task. Solving mazes can be easy with the right inductive bias. For example, matching the output dimensions to the input space, a model can perform binary classification at each location. Such a setup is amenable to machines by design, as they can learn iterative algorithmic solutions [37, 42], but this is not how humans solve mazes.

**Setup.**   The setup of our maze task deviates from the norm, specifically to necessitate the formation of an internal world model [43] by (1) requiring a direct sequence-of-actions output and (2) disallowing positional embeddings in the visual input. This requires a model to build its own spatial representation via observation (see Appendix D.6 for further discussion). We compare the CTM against LSTM and feed-forward (FF) baselines. For the results that follow, we trained a CTM, LSTMs (1, 2, and 3 layers), and a FF baseline to predict up to 100 steps down the path of $39 \times 39$ mazes, where predictions took the form of a sequence of classes for **l**eft, **r**ight, **u**p, **d**own, and **w**ait, using 'wait' to pad instances shorter than 100 steps. For the CTMs and LSTM baselines, we used 75 internal ticks, but LSTM stability issues meant that using 50 internal ticks yielded superior performance, so we report these too. In each case we used a automatic curriculum approach when training (see details in Appendix D.3). Appendices D.2 and D.4 detail hyperparameters for the CTM and baselines.

### 4.1   Results

The CTM significantly outperforms the baselines in solving these mazes, demonstrating superior trainability and generalization to longer paths (Figure 4). The FF model and LSTMs struggled to learn effectively or overfit (see Appendix D.5), whereas the CTM achieved high accuracy. This suggests that the CTM's architecture, particularly its use of neural dynamics and synchronization, is well-suited for tasks requiring robust internal state maintenance and planning.

### 4.2   Demonstrations and Generalization

Qualitative analysis shows the CTM methodically tracing paths (Figures 1a and 1b; supplementary video '*mazes.mp4*'), exhibiting emergent behavior such as continuing to explore paths beyond its training horizon. This suggests the CTM learns a general procedure rather than merely memorizing. Furthermore, the CTM, trained on $39 \times 39$ mazes, generalizes effectively to longer paths and larger $99 \times 99$ mazes (Figure 4c) by re-applying its learned policy, as shown in Figure 1c (see supplementary videos '*maze-large1.mp4*' to '*maze-large4.mp4*' for examples). Crucially, this CTM is not using any positional embedding, meaning that in order for it to follow a path through the maze it **must craft the**

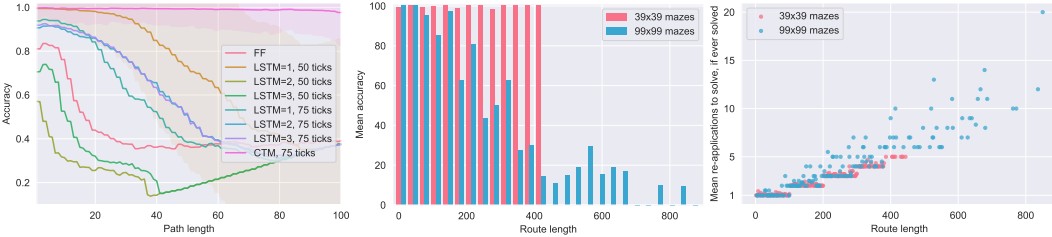

(a) Accuracies versus path length.     (b) Accuracies when generalizing     (c) Mean re-application count.

Figure 4: CTM versus baselines on 2D mazes. The CTM demonstrates superior trainability compared to baselines, yielding higher accuracy for longer paths. Using iterative re-applications, we show in (b) that the CTM can generalise to longer paths and bigger mazes. See Appendix D.5 for loss curves.

**cross-attention query by 'imagining' the future state of the maze**: a process known as 'episodic future thinking' [44] in humans. Appendix I discuss some of the emergent properties we observed.

# 5   ImageNet-1K Classification: Adaptive Processing and Emergent Dynamics

We evaluate the CTM on ImageNet-1K to understand its internal processing dynamics when trained to solve a standard classification task . We are not yet aiming for state-of-the-art accuracy (with 50 internal ticks and a ResNet-152 backbone: $72.47\%$ top-1, $89.89\%$ top-5 on uncropped data). Since the CTM uses new neural computation principles it would require a thorough hyperparameter search to find the optimal settings, and that is outside the scope of this work. Instead, we focus on *how* the CTM leverages neural dynamics (setup details in Appendix E.1) as a new mechanism for reasoning.

## 5.1   Adaptive Computation and Calibration

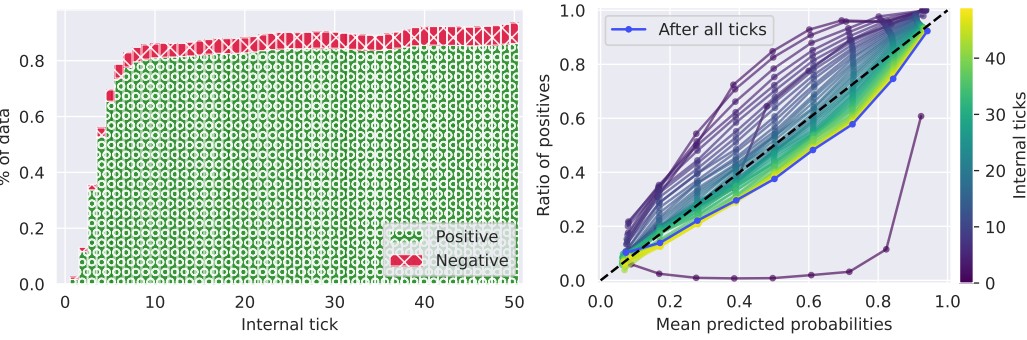

(a) Top-5 accuracies using a certainty threshold of 0.8.     (b) Calibration plot per internal tick.

Figure 5: ImageNet-1K results: (a) Native adaptive compute potential based on a 0.8 certainty threshold, showing performance expected at each internal tick. (b) Excellent model calibration when averaging probabilities up to each tick shown. See Appendix E.3 for further analysis.

The CTM exhibits adaptive computation: it can halt internal ticks based on prediction certainty. For instance, setting a certainty threshold of $0.8$ (Figure 5a) means that a user could halt compute for the majority of instances after fewer than 10 of 50 internal ticks. This is a consequence of internal recurrence couple with our novel loss function. The CTM also demonstrates excellent calibration (Figure 5b) as an emergent property of its iterative refinement process ( Appendix E.3).

## 5.2   Reasoning sequentially about static images

The CTM exhibits diverse temporal dynamics (Figure 2a), the synchronization of which is the representation with which it observes data and forms predictions. We show in Figure 2b how the CTM learns to 'look around' an image in order to gather information and make a prediction. It does this entirely without prompting or any guide, implementing computationally beneficial adaptive

compute in an intuitive fashion . This internal process can even manifest emergent phenomena like low-frequency traveling waves [45] across UMAP-projected neuron activations (see supplementary video '*umap.mp4*'). Unpacking every interesting facet of these attention map progressions is simply infeasible in a static form; we encourage viewing supplementary video '*imagenet.mp4*' for demonstrations of the CTM 'gazing' in a manner not quite entirely unlike how humans might look around images. Appendix E.4 has further demos and UMAP visualizations. These observations underscore that the CTM solves classification by leveraging an internal, dynamic reasoning process, a departure from typical feed-forward approaches.

# 6    Parity: Learning Sequential Algorithms and Interpretable Strategies

To test the CTM's ability to learn algorithmic procedures and develop interpretable strategies, we use a cumulative parity task: given a 64-length binary sequence, predict the parity at each position (Figure 6a). Unlike prior work focusing on final parity [18], our setup requires the model to output sequences at each internal tick, enabling us to examine how the full output evolves across ticks and throughout training. Setup details are in Appendix F.1.

## 6.1    Results and Learned Strategies

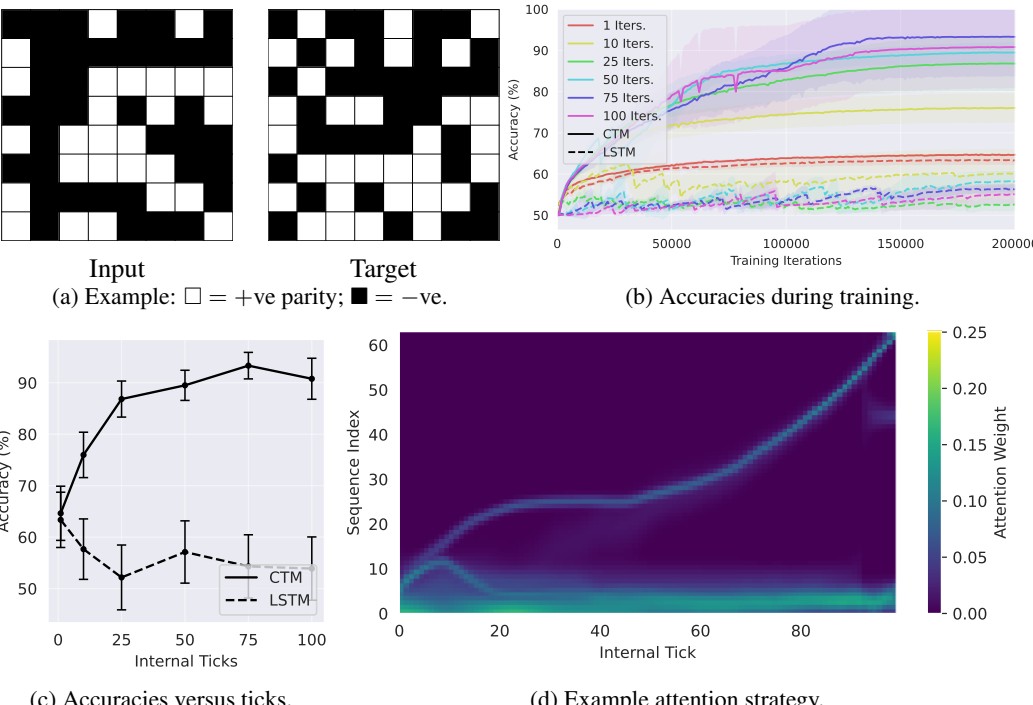

(a) Example: $\square$ = +ve parity; $\blacksquare$ = −ve.

(b) Accuracies during training.

(c) Accuracies versus ticks.

(d) Example attention strategy.

Figure 6: CTM performance on the parity task: (a) example; (b) training accuracy comparisons; (c) Impact of internal ticks on accuracy; and (d) an example showing how this CTM uses at least one attention head to scan the input sequence from start to end. Error bars (b, shaded) represent 1 standard deviation over seeded runs; Appendix F.2 discusses the implications of seed variations.

The CTM's accuracy improves with more internal ticks, significantly outperforming parameter-matched LSTMs, which struggled with stability and performance (Figure 6b). CTMs with 75 and 100 ticks could achieve perfect accuracy in some seeded runs. Figure 6d shows how the attention shifts over the input data, and Figure 7 shows a specific demonstration (4 of 8 attention heads), revealing a distinct and interpretable strategy. Which specific 'style' of solution depends on the configuration and seed, so we show other examples and analyses in Appendix F.2). Crucially, this experiment demonstrates that the CTM can learn to form and follow an internal strategy for an algorithmic task.

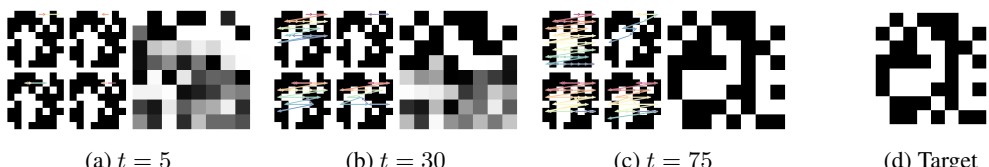

(a) $t = 5$     (b) $t = 30$     (c) $t = 75$     (d) Target

Figure 7: Determining parity: (a, b, c) are the trajectories of the argmax of attention for 4 heads and the corresponding prediction at different internal ticks, and (d) is the target (perfectly predicted here). See supplementary material '*parity.mp4*' for video format.

## 7 Other Experiments and Analyses

We also evaluated the CTM in a number of other settings in order to probe its functionality and versatility. Owing to space constraints, we provide the details of these additional experiments in the appendices (referenced below). In summary, these additional experiments investigated:

**CIFAR-10 Classification Compared to Humans** (Appendix G.1): The CTM, feed-forward, and LSTM baselines were evaluated on CIFAR-10, with results compared against human data for difficulty and uncertainty. The CTM demonstrated good model calibration and alignment with humans.

**CIFAR-100 Ablation Studies** (Appendix G.2): We investigated the impact of model width and the number of internal ticks. We found that the diversity of neural activity are functions of these. Wider models tended to exhibit more varied neural dynamics. Using more internal ticks allowed the CTM to engage in extended processing, sometimes revealing distinct computational phases.

**Neuron-Level Models and Synchronization Ablations** (Appendix G.3): We compared the CTM to parameter-matched variants without NLMs and without synchronization, as well as an LSTM with synchronization. The results show that the combination of neuron-level models and synchronization as a representation is key to the success of the CTM.

**Sorting Real Numbers** (Appendix G.4): The CTM was tasked with sorting sequences of 30 real numbers, outputting sorted indices sequentially using a Connectionist Temporal Classification (CTC) loss [46]. This experiment showed that the CTM could learn an algorithmic sorting procedure and exhibited adaptive computation by varying its internal processing duration ("wait times") based on characteristics of the input sequence, such as the difference between successive values.

**Q&A MNIST** (Appendix G.5): In this task, the CTM processed sequences of MNIST digits followed by index and operator embeddings to perform multi-step modular arithmetic. This investigation highlighted the CTM's capacity for memory and retrieval, using its synchronization mechanism to recall digit information beyond the immediate history window of individual neuron-level models, and to generalize to longer computational sequences than seen during training.

**Reinforcement Learning** (Appendix G.6): The CTM was adapted for reinforcement learning in several partially observable Markov decision processes (POMDPs), including classic control (CartPole, Acrobot) and grid-world navigation (MiniGrid Four Rooms). This demonstrated the CTM's applicability to sequential decision-making in continuous interaction settings, where it achieved performance comparable to LSTM baselines while developing richer internal state dynamics.

## 8 Discussion and Conclusion

The Continuous Thought Machine (CTM) represents a new perspective, where the temporal dynamics of neural activity are central to artificial cognition. Its core innovations—neuron-level models and synchronization as a latent representation—effectively enable it to both unfold and leverage neural dynamics to solve problems. We showed in this work that such an approach is not only feasible but also leads to unique computational capabilities and emergent properties.

Our experiments demonstrate that the CTM can effectively solve challenging tasks. We trained a CTM to observe, plan, and implement routes through 2D mazes using a setup that necessitated the

formation of an internal world model. On ImageNet, the CTM exhibited native adaptive computation, naturally tailoring its processing time to input difficulty, and achieved strong calibration—a desirable property often requiring specialized techniques. On algorithmic tasks like parity checking, the CTM developed interpretable, sequential problem-solving strategies. Notably, the core architecture remained consistent across tasks, highlighting its robustness.

The CTM's NLMs are inspired by the complexity of biological neurons, but are implemented with a level of abstraction appropriate for modern deep learning . The direct use of neural synchronization as a representation is, to our knowledge, a novel approach at this scale, offering benefits such as a high-cardinality representational space and the potential to capture the temporal aspects of 'thought'. While traditional deep learning has abstracted away neural timing for computational efficiency, the CTM shows that reintroducing such dynamics in a structured way can unlock new functionalities.

**Limitations.**   The CTM uses an internal sequence, meaning training times are extended. NLMs also increase parameter counts compared to standard activation functions, but also provide a new avenue for scaling. The experiments in this paper are preliminary and not intended to beat state-of-the-art models tailored for performance, therefore a limitation of this paper is its relatively limited depth of comparison since we favored breadth to investigate the CTM's internal functionality.

**Future Work.**   We plan to apply the CTM to language modeling, self-supervised video understanding, lifelong-learning, biologically-inspired memory and plasticity, multi-modal systems, and more. We believe that, conceptually, synchronization representations have high widespread potential.

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

## Appendices

The following appendices provide further details about the architecture, experimental setup, and describe additional experiments we ran while undertaking this research. Specifically, these appendices are structured as follows: we begin with a glossary of key terms (Appendix A), followed by a collection of code listings to support our main descriptions (Appendix B). We then present a more detailed explanation of the core method (Appendix C), including the synapse model (Appendix C.1) and our approach to neuron sampling for synchronization (Appendix C.2). Next, we outline the experimental setups for the maze (Appendix D), ImageNet-1K (Appendix E), and parity tasks (Appendix F). This is followed by in-depth descriptions of further experiments (Appendix G), including results on CIFAR-10 versus humans (Appendix G.1), ablations using CIFAR-100 (Appendix G.2), ablating the use of NLMs and synchronization as a representation in a maze task (Appendix G.3) , sorting (Appendix G.4), question-answering and memory (Appendix G.5), and reinforcement learning (Appendix G.6). We provide details for our fast recursive formulation to computing synchronization (Appendix H). We conclude by providing insights into some of the emergent phenomena we observed while training or working with the CTM in Appendix I.

## A  Glossary

We provide a glossary containing the terminology and symbols used throughout this paper, in Table 1 and Table 2 respectively.

| Term | Description |
|---|---|
| Internal tick | One step of internal computation. |
| Memory length | Length of rolling history of pre-activations, updated in a rolling FIFO fashion |
| Synapse model | Recurrent model that takes $\mathbf{z}^t$ and $\mathbf{o}^t$ as input to produce pre-activations, $\mathbf{a}^t$. |
| Pre-activations | Output of recurrent synapse model, input to NLMs. |
| Post-activations | Output of NLMs, neuron states at time $t$. |
| (Pre/Post) activation history | Sequentially ordered history of activations over time. |
| Neuron-Level Model (NLM) | Per-neuron MLP over pre-activation history. |
| Synchronization | Dot product of post-activation histories. |
| Self-pair | Diagonal synchronization matrix entries $(i, i)$. |
| Action synchronization | Synchronization representation for attention queries. |
| Output synchronization | Synchronization representation for predictions. |
| Decay $r_{ij}$ | Learnable time decay for synchronization (action or output). |
| Feature extractor | Task-specific input encoder (e.g., ResNet). |
| Attention output | Output after cross-attention using queries, $\mathbf{q}^t$, computed from action synchronization, and keys/values from data. |

Table 1: Glossary of terms.

## B  Listings

In this section, we describe the core aspects of the CTM using pseudocode. Specifically, Listing 1 gives a simplified overview of the CTM code. Listing 2 showcases how the NLMs can be written using Einstein summation. Listing 3 shows how we compute synchronization. Listing 4 shows how the certainty-based loss described in Section 3.5 is computed.

## C  Method details

### C.1  Synapse models

Figure 8 show the synapse model which is the recurrent structure that shares information across neurons in the CTM. It is implemented by choosing a depth of $k$ (always even), where each subsequent

| Symbol | Meaning |
|--------|---------|
| $T$ | Number of internal ticks |
| $M$ | Memory length |
| $d_{\text{model}}$ | Dimensionality of latent state in the CTM |
| $d_{\text{input}}$ | Dimensionality of attention output |
| $d_{\text{hidden}}$ | Hidden size in each neuron's private MLP (NLM) |
| $k$ | Depth of the synapse MLP or U-Net |
| $p_{\text{dropout}}$ | Dropout probability in the synapse model |
| $n_{\text{heads}}$ | Number of heads in multi-head attention |
| $J_{\text{action}}$ | Number of neurons used for action synchronization |
| $J_{\text{out}}$ | Number of neurons used for output synchronization |
| $D_{\text{action}}$ | Dimensionality of action synchronization vector |
| $D_{\text{out}}$ | Dimensionality of output synchronization vector |
| $n_{\text{self}}$ | Number of self-pairs $(i, i)$ used in synchronization sampling |
| $r_{ij}$ | Learnable decay parameter for synchronization between neuron $i$ and $j$ |
| $\mathbf{S}^t$ | Full synchronization matrix at internal tick $t$ |
| $\mathbf{q}^t$ | Query vector projected from action synchronization |
| $\mathbf{y}^t$ | Output vector (e.g., logits) projected from output synchronization |

Table 2: Glossary of symbols.

layer width is chosen to linearly reduce the dimensionality until a width of 16 is reached, and then increase thereafter, using skip connections to retain information. The synapse model also takes $\mathbf{o}^t$ (the output of attention) as input.

### C.2   Sampling synchronization neurons

The CTM operates using recurrence on a latent representation, $\mathbf{z}$, that is $D$-dimensional. $\mathbf{z}$ unfolds over time and the synchronization between *some chosen neurons* form the new kind of representation that the CTM enables.

There are $\frac{D \times (D+1)}{2}$ unique pairs of neurons, making for a substantially **larger set of neuron synchronization pairs** than there are neurons themselves. This motivates the need for a selection process. Over the development of the CTM we came up with three approaches to selecting neurons:

1. **Dense pairing**: in this setup we select $J$ neurons and compute synchronization for every possible $(i, j)$ pair of the $J$ neurons. For $\mathbf{S}_{\text{out}}^t$ we choose $J_{\text{out}}$ neurons and for $\mathbf{S}_{\text{action}}^t$ we choose non-overlapping $J_{\text{action}}$ neurons. Selecting $J_{\text{out}}$ neurons for dense pairing results in an output synchronization representation of $D_{\text{out}} = \frac{J_{\text{out}} \times (J_{\text{out}}+1)}{2}$, and similarly for the action representation.

   This approach essentially creates a strong bottleneck where all gradients must flow through the selected neurons, which can be advantageous for some tasks.

2. **Semi-dense pairing**: in this setup we open up the aforementioned bottleneck twofold by selecting two different subsets, $J_1$ and $J_2$, such that the left neurons of the synchronization dot product, $i$, are taken from $J_1$ and the right neurons, $j$, are taken from $J_2$. The same dense computation as before is then applied.

   The bottleneck width in this case is $2\times$ as wide as before. Output and action selections are, once more, not overlapping.

3. **Random pairing**: in this setup we randomly select $D_{\text{out}}$ or $D_{\text{action}}$ pairs of neurons and compute the synchronization between each pair as opposed to doing so densely between all selected neurons. We also intentionally compute the $(i, i)$ dot products between $n_{\text{self}}$ neurons in each case in order to ensure that the CTM could recover a snapshot representation if it wanted to.

   This opens up the bottleneck much more than before. We allow overlapping selections in this case.

```
################# DEFINITIONS  (hyper parameters not shown for simplicity) #################
# Backbone can be, e.g., a ResNet for images
backbone = FeatureEncoder()
# Q and KV projectors, and standard attention module
q_projector, kv_projector = Linear(), Linear()
attn = MultiHeadAttention()
# Synapse model can be linear or MLP or U-NET
synapses = MLP()
# Neuron level models (see Listing 2)
neuron_level_models = NLMS()
# Output projector from synchronisation (see Listing 3)
output_proj = Linear()
# Initialise pre-activations and z as learnable parameters
# D is the model width
z_init = Parameter(size=(D))
# M is the neuron memory length
pre_acts_history_init = Parameter(size=(D, M))

##################################### MODEL LOGIC #####################################
# Featurise inputs with backbone and compute KV tokens
kv = kv_projector(backbone(inputs))
# In each minibatch, initialise the learnable pre_act_history
pre_acts_history = pre_acts_history_init.unsqueeze(0).repeat(B, dim=0)  # (B, D, M)
# And start the post_acts_history with the learnable z_init
post_acts_history = [z_init.unsqueeze(0).repeat(B, dim=0)]
outputs_history = []
# Get initial action synchronisation to query data
synch_a = compute_synch(post_acts_history, type="action")
# Other initialisations, including learnable start histories and first pre-attn round
for step in range(n_thought_steps):
    # Project attention query off synchronisation
    q = q_projector(synch_a)
    attn_out = attn(q, kv, kv)
    # Concatenate attention output and process via synapses
    pre_acts = synapses(concat((attn_out, z)))
    # Keep history of pre-activations. This is a FIFO structure:
    pre_acts_history = concat((pre_acts_history[:, :, :-1], pre_acts), dim=-1)
    # Compute post-activations using histories (see Listing 2)
    z = neuron_level_models(pre_acts_history)
    post_acts_history.append(z)
    # Compute synchronisations (see Listing 3)
    synch_a = compute_synch(post_acts_history, type="action")
    synch_o = compute_synch(post_acts_history, type="output")
    # Projecte prediction/output off synchronisation
    outputs_history.append(output_proj(synch_o))
# Return outputs per thought step for loss function
return outputs_history
```

Listing 1: Simplified overview of the CTM code. Features are encoded using a backbone (e.g., ResNet layers for images), data is attended to by ⑨ projecting a query from neural synchronization, information is shared across neurons using an ① MLP synapse model to produce pre-activations, ③ private neuron-level models are applied to a ② tracked history of pre-activations (see Listing 2), synchronization is computed from a ⑤ tracked history of post-activations (see Listing 3), and outputs are ⑨ projected off of synchronization.

# D  2D Mazes

## D.1  Dataset

We used the maze-dataset repository to generate mazes for this work. We generated mazes of size $19 \times 19$, $39 \times 39$, and $99 \times 99$. In each case we generated 50000 mazes and split them into train sets of size 45000 and test sets of size 5000. We used the $39 \times 39$ for training in this technical report and tested generalization on the $99 \times 99$. We provide all three maze datasets[3] in the CTM code repository, made available upon publication.

## D.2  Architecture details

We used the following hyperparameters:

- $39 \times 39$ mazes and a ResNet-34 backbone, where the keys and values were taken as features after the second hyper-block, resulting in a down-sample to $10 \times 10$
- $D = 2048$ (the width of $\mathbf{z}^t$ and $\mathbf{a}^t$)

---

[3]We found the $19 \times 19$ beneficial for debugging, hence we provide it too.

```
# Initialisations
weights_1 = Parameter(shape=(M, d_hidden, d_model))
bias_1 = zeros(shape=(1, d_hidden, d_model))
weights_2 = Parameter(shape(d_hidden, d_model))
bias_2 = zeros(shape=(1, d_model))
# Forward pass
# b=batch, M=memory, d=d_model, h=d_hidden
# inputs are shape (b, d, M)
inputs = pre_acts_history[-M:]
out = einsum('bdM,Mhd->bdh', inputs, weights_1) + bias_1
out = einsum('bdh,hd->bd', out, weights_2) + bias_2
```

Listing 2: Neuron-level models: ③. Using einsum greatly simplifies and speeds up the application of neuron-level models as their outputs can be computed in parallel. The first einsum computes a h-dimension latent for each neuron from the (truncated to $M$ most recent) incoming history, ②. The second einsum then computes the single activation per-neuron (ignore the '1' dimension here for simplicity).

```
# AT INITIALISATION:
# Pre choose D_chosen neuron pairs from D total neurons
# D_chosen can be D_out or D_action
# Other neuron selection strategies exist, but here we show random selection
idxs_left = randint(low=0, high=D, size=D_chosen)
idxs_right = randint(low=0, high=D, size=D_chosen)  # can overlap
# Define learnable exponential decay scaling factors per neuron pair
r = Parameter(zeros(1, D_chosen, 1))
# INITIALISATION OVER.
# IN FORWARD PASS:
S = stack(post_acts_history, 1)  # S is of shape [B, T=history length]
# decay BACK in time
t_back = range(T-1, -1, -1).reshape(1, T, 1)
# Compute per NEURON PAIR exponential decays, and expand over D_chosen
exp_decay = exp(-t_back * r).expand(1, T, D_chosen)
# Compute weighted inner dot products using differet subsets of neurons
S_multiplied = S[:,:,idxs_left] * exp_decay * S[:,:,idxs_right]  # [B, T, D_chosen]
# Sum over the free T dimension and normalise by sqrt of AUC of decays
synch_representation = (S_multiplied).sum(1)/sqrt(exp_decay.sum(1))  # [B, D_chosen]
```

Listing 3: Neural synchronization to create the latent representations used in Listing 1. Careful reshaping and broadcasting enables the use of per neuron-pair learnable exponential decays for synchronization, which lets the CTM learn complex timing dependencies. The decay parameters are initialized as zeros (i.e., no decay). This process is repeated for output and action (see Section 3.4.1). In practice we use a recursive approach that greatly reduces compute overhead; see Appendix H.

- $k = 16$ (synapse depth, $8$ layers down and $8$ layers up)
- $d_{\text{input}} = 512$ (the width of attention output, $\mathbf{o}^t$)
- $n_{\text{heads}} = 16$
- Dense pairing for neuron selection (see Appendix C.2)
- $J_{\text{out}} = 32$ (width of $\mathbf{S}_{\text{out}}^t$ synchronization representation)
- $J_{\text{action}} = 32$ (width of $\mathbf{S}_{\text{action}}^t$ synchronization representation)
- $T = 75$ (internal ticks)
- $M = 25$ (FIFO rolling memory input to NLMs)
- $d_{\text{hidden}} = 32$ (width of MLPs inside NLMs)
- $p_{\text{dropout}} = 0.1$ (dropout probability for synapse model)
- No positional embedding

We used the following settings for optimization:

- Trained using a batch size of 64 on 1 H100 Nvidia GPU
- 1000000 iterations for training using AdamW [47]
- A learning rate of 1e-4 with a linear warmup of 10000 iterations and decaying to zero using a cosine annealing learning rate scheduler

```
def ctm_loss(logits, targets):
    B, C, T = logits.shape
    # B=minibatch size, C=classes, T=thought steps
    # Targets shape: [B]
    # Compute certainties as 1 - normalised entropy
    p = F.softmax(logits, 1)
    log_p = torch.log_softmax(logits, 1)
    entropy = -torch.sum(p * log_p, dim=1)
    max_entropy = torch.log(C)
    certainties = 1 - (entropy / max_entropy)
    # Certainties shape: [B, T]
    # Expand targets over thought steps
    targets_exp = torch.repeat_interleave(targets.unsqueeze(-1), T, -1)
    # Loss function could be other things, but we use cross entropy without reduction
    loss_fn = nn.CrossEntropyLoss(reduction='none')
    # Losses are of shape [B, T]
    losses = loss_fn(predictions, targets_exp)
    # Get indices of lowest loss thought steps for each item in the minibatch
    lowest_idx = losses.argmin(-1)
    # Get indices of most certain steps for each item in the minibatch
    certain_idx = certainties.argmax(-1)
    loss = (losses[:, lowest_idx] + losses[:, certain_idx])/2
    return loss.mean()
```

Listing 4: CTM loss function, enabling the CTM to be flexible regarding the number of internal ticks used for any given data point.

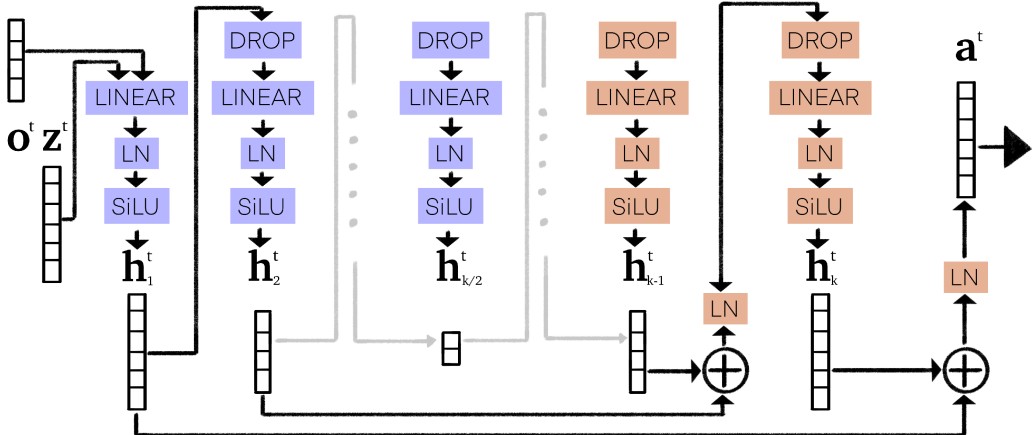

Figure 8: Overview of UNET style 'synapse' recurrent models. $\mathbf{z}^t$ are the post-activations from the previous step, $\mathbf{o}^t$ are the attention outputs from observing data, and the synapse model yields $\mathbf{a}^t$ pre-activations for the NLMs to process. The UNET structure is set up so that the innermost bottleneck layer is 16 units wide, with linear scaling for each layer in between. Skip connections with layer-norm implement the classic UNET structure to maintain information. Layers that produce lower dimensionality are shown in blue, while layers that produce higher dimensionality are shown in orange.

- No weight decay

This resulted in a model with 31,998,330 parameters.

### D.3 Maze curriculum

We adapted the loss function for solving mazes to include a curriculum element. Before computing $t_1$ and $t_2$ for the loss in Equation (11) we first altered the **loss at each internal tick** to only account for those steps in the maze that were correctly predicted with plus 5 additional steps along the path. This effectively lets the model (CTM or LSTM baselines) slowly learn to solve the maze from start to finish. Listing 5 shows how this is implemented when computing the loss and is used for both CTM and LSTM training.

```python
def image_classification_loss(predictions, certainties, targets, use_most_certain=True):
    """
    Computes the maze loss with auto-extending cirriculum.

    Predictions are of shape: (B, class, internal_ticks),
    Certainties are of shape: (B, 2, internal_ticks),
        where the inside dimension (2) is [normalised_entropy, 1-normalised_entropy]
    Targets are of shape: [B]

    use_most_certain will select either the most certain point or the final point.
    """
    targets_expanded = torch.repeat_interleave(targets.unsqueeze(-1), predictions.size(-1), -1)
    # Losses are of shape [B, internal_ticks]
    losses = nn.CrossEntropyLoss(reduction='none')(predictions, targets_expanded)

    loss_index_1 = losses.argmin(dim=1)
    loss_index_2 = certainties[:,1].argmax(-1)
    if not use_most_certain:  # Revert to final loss if set
        loss_index_2[:] = -1

    batch_indexer = torch.arange(predictions.size(0), device=predictions.device)
    loss_minimum_ce = losses[batch_indexer, loss_index_1].mean()
    loss_selected = losses[batch_indexer, loss_index_2].mean()

    loss = (loss_minimum_ce + loss_selected)/2
    return loss
```

Listing 5: Loss function implementation of Section 3.5 for maze route prediction, including an auto curriculum approach for both CTM and LSTM.

## D.4 Baselines details

We tested a number of LSTM baselines for solving this task, but struggled with stability during training (see Figure 4), particularly beyond a single LSTM layer or with a higher number of internal ticks. Hence we tested three LSTM configurations of depths 1,2, and 3. For each model we tested with 75 internal ticks to match the CTM, and 50 internal ticks for stability. We also tested a feed-forward model, projecting the feature space (before average pooling) into a hidden layer of the same width as the CTM, yielding a slightly higher parameter count. We kept all hyperparameters constant, yielding the following setups:

- **LSTM**, 1 layer, $T = 50$ and $T = 75$: 42,298,688
- **LSTM**, 2 layers, $T = 50$ and $T = 75$: 75,869,504 parameters
- **LSTM**, 3 layers, $T = 50$ and $T = 75$: 109,440,320 parameters
- **Feed-forward**, with a hidden layer width of 2048 (and GLU activation thereon): 54,797,632 parameters

## D.5 Maze loss curves

Figure 9 gives the loss curves for the maze solving models in Section 4.1, showing how the CTM is more stable and performant when trained on this task.

## D.6 Discussion: the need for a world model and cognitive map

Internal models of the world and cognitive maps represent crucial aspects of intelligent systems [43, 48, 49]. In this case, we consider a world model to be an internal representation of the external environment, encapsulating an agent's knowledge about the world's structure, its dynamics, and its actionable place therein. A good world model should enable an agent to reason about the world, plan, and predict the consequence of its actions. Cognitive maps [48] specifically focus on spatial relationships and navigation. The ability to construct and utilize these internal representations is a strong indicator, and arguably a prerequisite, for sophisticated intelligence. The notion of 'episodic future thinking' [44] is even considered a hallmark feature of human intelligence. An agent devoid of a world model would be limited to reactive behaviors. Similarly, lacking a cognitive map would severely restrict an agent's ability to navigate and interact effectively within complex spatial environments. Therefore, the presence and sophistication of world models and cognitive maps can serve as a benchmark for evaluating intelligence.

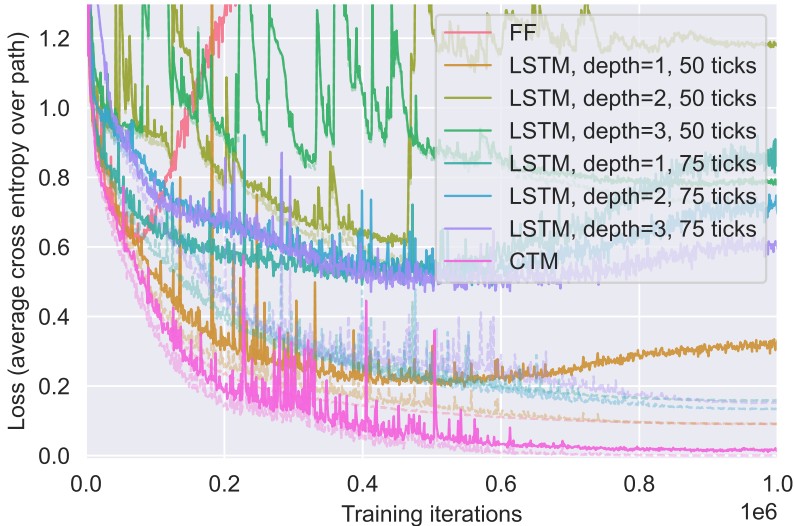

Figure 9: Loss curves when training the CTM and baselines.

To this end, we designed the maze task such that it would require a good internal world model to solve. This was achieved by (1) requiring the model to output a route directly, as opposed to solving the maze with a local algorithm [37], and (2) forgoing any positional embedding in the image representation, meaning that the model must build its own spatial cognitive map in order to solve the task [48]. Indeed, we saw that the NLMs and synchronization components of the CTM enables it to solve our 2D maze task, far surpassing the best baselines we trained. Our results suggest that the CTM is more capable of building and utilizing an internal model of its environment.

# E    ImageNet-1K

This section provides additional details and results for the ImageNet-1K experiments.

## E.1    Architecture details

We used a constrained version of the classic ResNet architecture [41] for this task, which we adapted from `https://github.com/huyvnphan/PyTorch_CIFAR10`. It differs from the standard implementation for ImageNet in that the first convolution is constrained to use a kernel size of $3 \times 3$ as opposed to $7 \times 7$. We used a ResNet-152 structure and took the output prior to the final average pooling and projection to class logits. We used input images of size $224 \times 224$ which yielded $14 \times 14$ features for the keys and values used in cross attention.

We used the following hyperparameters:

- $D = 4096$ (the width of $\mathbf{z}^t$ and $\mathbf{a}^t$)
- $k = 16$ (synapse depth, 8 layers down and 8 layers up)
- $d_{\text{input}} = 1024$ (the width of attention output, $\mathbf{o}^t$)
- $n_{\text{heads}} = 16$
- Random pairing for neuron selection (see Appendix C.2)
- $D_{\text{out}} = 8196$ (width of $\mathbf{S}^t_{\text{out}}$ synchronization representation)
- $D_{\text{action}} = 2048$ (width of $\mathbf{S}^t_{\text{action}}$ synchronization representation)
- $n_{\text{self}} = 32$ (for recovering a snapshot representation)
- $T = 50$ (internal ticks)
- $M = 25$ (FIFO rolling memory input to NLMs)
- $d_{\text{hidden}} = 64$ (width of MLPs inside NLMs)

- $p_{\text{dropout}} = 0.2$ (dropout probability for synapse model)
- No positional embedding

We used the following settings for optimization:

- Trained using a batch size of 64 across 8 H100 Nvidia GPUs
- 500000 iterations for training, using a custom sampling such that each minibatch was sampled with possible replacement
- AdamW [47]
- A learning rate of 5e-4 with a linear warmup of 10000 iterations and decaying to zero using a cosine annealing learning rate scheduler
- Gradient norm clipping set to a norm of 20
- No weight decay

## E.2  Loss function

Listing 6 shows the python code for the image classification loss function used to train the CTM on ImageNet-1K. This accompanies the loss defined in Section 3.5.

```python
def image_classification_loss(predictions, certainties, targets, use_most_certain=True):
    """
    Computes the maze loss with auto-extending cirriculum.

    Predictions are of shape: (B, class, internal_ticks),
    Certainties are of shape: (B, 2, internal_ticks),
        where the inside dimension (2) is [normalised_entropy, 1-normalised_entropy]
    Targets are of shape: [B]

    use_most_certain will select either the most certain point or the final point.
    """
    targets_expanded = torch.repeat_interleave(targets.unsqueeze(-1), predictions.size(-1), -1)
    # Losses are of shape [B, internal_ticks]
    losses = nn.CrossEntropyLoss(reduction='none')(predictions, targets_expanded)

    loss_index_1 = losses.argmin(dim=1)
    loss_index_2 = certainties[:,1].argmax(-1)
    if not use_most_certain:  # Revert to final loss if set
        loss_index_2[:] = -1

    batch_indexer = torch.arange(predictions.size(0), device=predictions.device)
    loss_minimum_ce = losses[batch_indexer, loss_index_1].mean()
    loss_selected = losses[batch_indexer, loss_index_2].mean()

    loss = (loss_minimum_ce + loss_selected)/2
    return loss
```

Listing 6: Loss function implementation of Section 3.5 for standard classification tasks, used for ImageNet-1K.

## E.3  Further analysis

**A note on how we compute calibration.**  For Figure 5 we showed the calibration plots per internal tick. To compute these we averaged the probability of the predicted class (at a given tick) over all ticks preceding that. This approach is aligned with the way that the CTM reasons and builds up its predictions. For example, an uncertain prediction (e.g., Figure 15) would have low certainty over all ticks, and typically exhibit only gradual increase in certainty, whereas a certain prediction (e.g., Figure 2b) would show a sharp increase in certainty early on, yielding an average predictive probability close to 1 as thought progressed.

We also give accuracies over internal ticks in Figure 10a for four styles of predictions:

1. **Instant**: by taking the prediction at a given internal tick.
2. **Most certain**: by taking the prediction at the most certain tick up to a given tick.
3. **Average logits**: taking the average logits up to a given tick.

4. **Average logits weighted by certainty**: by first re-weighting logits prior to aggregating up to a given tick.

This shows how the process of aggregating or using the predictions from the CTM is non-trivial. We hope that this opens avenues for future research.

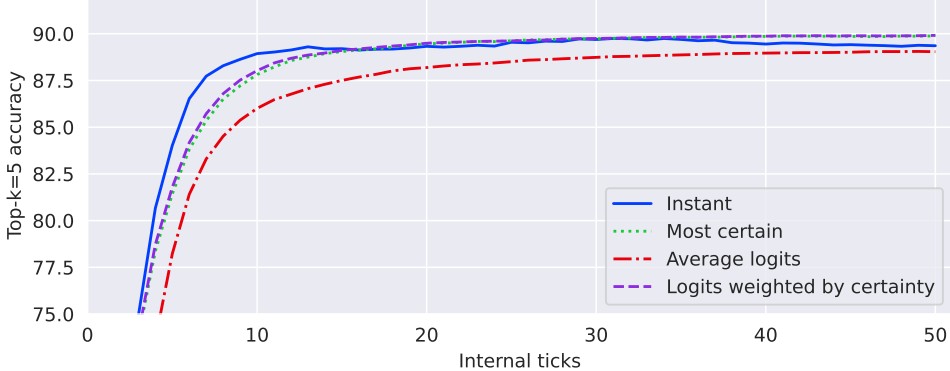

(a) Accuracy vs. steps for different prediction mechanisms.

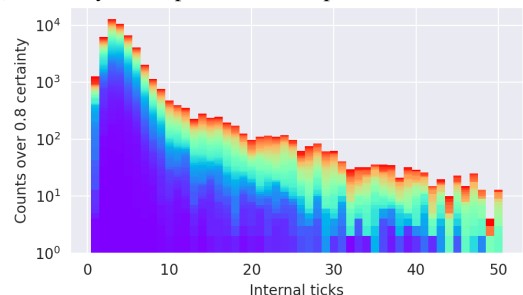

(b) Histogram of certainty of at least 0.8.

Figure 10: Exploration of the performance and utility of the CTM, showing the relationship between internal ticks and top-5 ImageNet-1K accuracy. In (a) we show the accuracy versus internal ticks when determining the output prediction in 4 different ways, showing how taking the prediction at a given internal tick is sensible until approximately 15 steps, where it becomes better to consider the certainty as a measure of success. In (b) we show a histogram of data counts exceeding a certainty of 0.8 for each internal tick; color denotes class indices. In (c) we show calibration plots, where predicted probabilities for the CTM are considered to be the average probability up to a given internal tick, showing how this results in good model calibration.

In Figure 10b we show the distribution of classes (in different colours) over each internal tick.

### E.4 Additional demonstrations

We showcase a series of additional demonstrations of CTMs classifying images from ImageNet in Figures 12 to 15. We encourage the reader to view similar demonstrations as videos (we will provide links upon publication; see supplementary video '*imagenet.mp4*')

To visualize the dynamics of the neuron activations we used UMAP [50] to project the neurons to a 2D feature space. We used the history of post-activations for 200 different ImageNet images as the high dimensional inputs to UMAP. Specifically, this input was $200 \times T = 200 \times 5 = 1000$ dimensional. The resulting 2D projection assigns each neuron to a position in space determined by its activation 'profile' – its response pattern both over time and over multiple stimuli. Visualizing this mapping over internal ticks reveals low frequency structures propagating across the feature space. We show snapshots of this visualization over the internal ticks in Figure 11, noting that these visualizations are best viewed in video form (see supplementary video '*umap.mp4*').

Importantly, the CTM generates this structure in an emergent fashion, without any explicit driving signal. Analogous phenomena occur in networks of Kuramoto oscillators [51]; in our case, waves

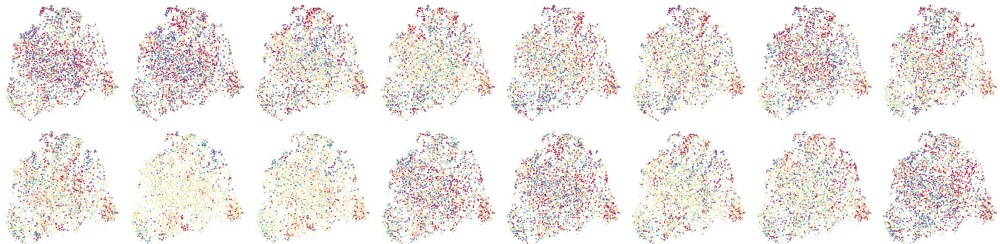

Figure 11: Observation of the neurons in the CTM as it observes and thinks about an image. The colors indicate activations that range from low (blue) to high (red). We show the progression of the neural activity over internal ticks from top left to bottom right. Upon careful inspection, one can discover clear structures at multiple scales. This visualization is best viewed in video form. (See supplementary video '*umap.mp4*').

propagate across a learned feature map in an all-to-all network. Concurrent work also explores explicitly encoding traveling waves for long-range communication [52]. We do not assign functional meaning to these observed waves but highlight their distinct presence during the CTM's thought process.

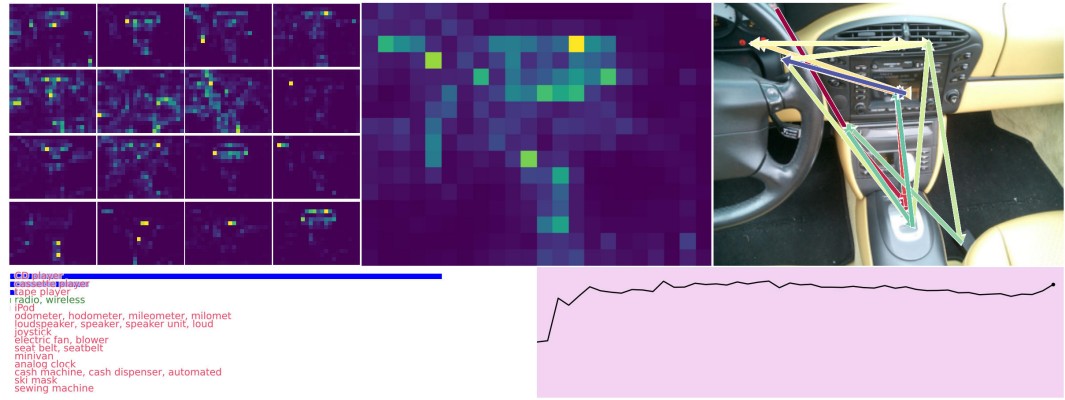

Figure 12: Validation image index 1235, showing incorrect and uncertain prediction.

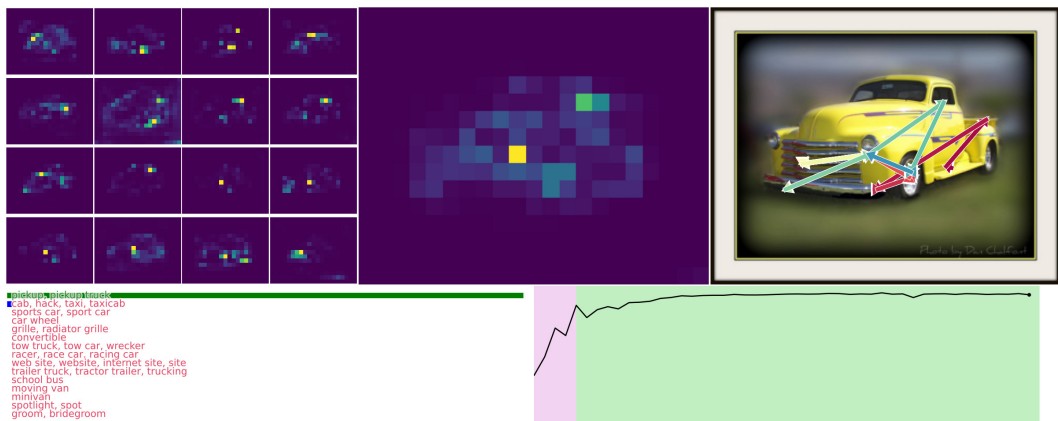

Figure 13: Validation image index 15971, showing correct prediction and plausible 2nd most probable class.

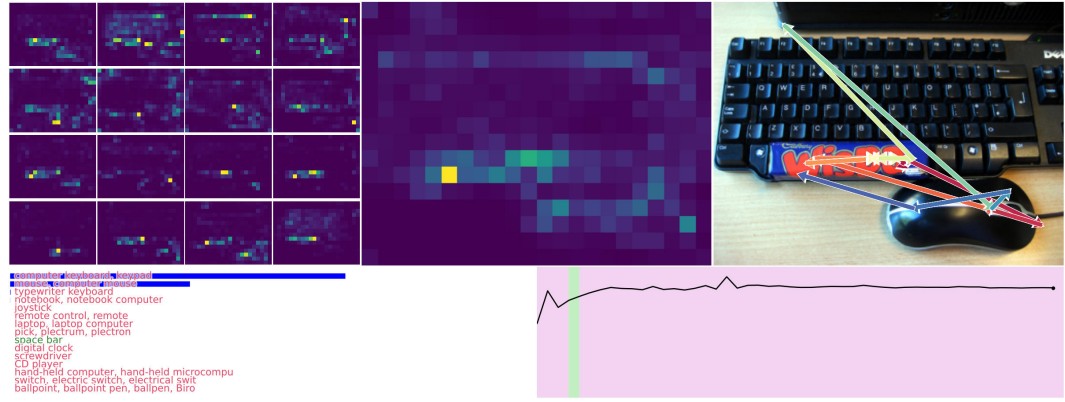

Figure 14: Validation image index 21202, incorrect prediction after passing by correct prediction, showing 'over-thinking'.

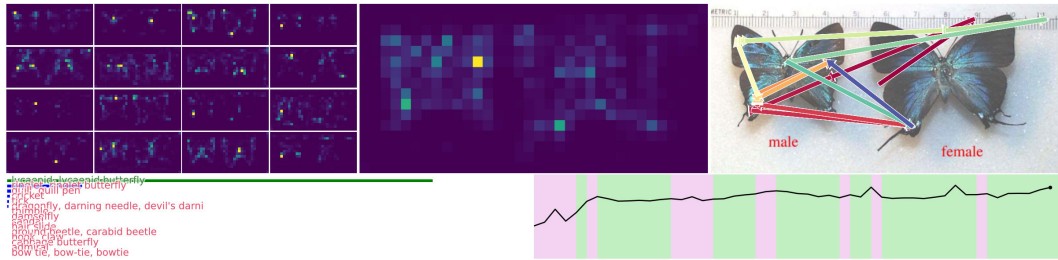

Figure 15: Validation image index 39275, correct but uncertain prediction.

# F  Parity

## F.1  Task Details

The parity of a binary sequence is given by the sign of the product of its elements. When processing a sequence element by element, an RNN could conceivably compute parity by maintaining an internal state, flipping an internal 'switch' whenever a negative number is encountered. However, if the entire sequence is provided simultaneously, the task increases in difficulty due to the increasing number of distinct patterns in the input. Previous work [18] has addressed this challenge using recurrent models, which can learn sequential algorithms for statically presented data. Computing parity, as posed in this manner, is well-suited for testing the capabilities of the CTM.

We apply the CTM to the task of computing the parity of a 64-length sequence containing the values 1 and -1 at random positions. Unlike [18], we set up the task such that the model computes the cumulative parity at every index of the sequence, not just the final parity. An example is shown in Figure 6a. The values -1 and 1 are embedded as learnable vectors combined with positional embeddings, using attention to ingest input data. We train the CTM with the loss function described in Section 3.5. As a baseline we also trained an LSTM, but set $t_2$ to be the final iteration since this gave the best results and stability for LSTM training.

## F.2  Additional Results and Analysis

In this section, we describe some additional results for the parity experiments.

**The CTM learns a sequential algorithm.**    To analyze how the CTM learns to solve the parity task, Figure 16 shows the accuracy for each of the 64 elements in the input sequence at different stages of training, for three different internal tick configurations. The models first learn to predict the parity of the initial elements, and as training proceeds, learn to predict later and later positions. With more internal ticks, the model can accurately predict more elements in the target sequence.

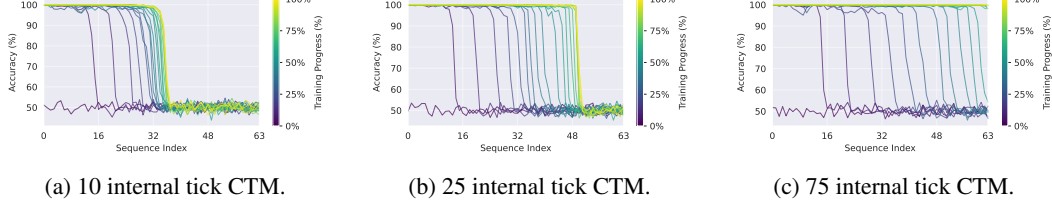

(a) 10 internal tick CTM.    (b) 25 internal tick CTM.    (c) 75 internal tick CTM.

Figure 16: Accuracy across the 64-element sequence at different training stages (indicated by color) for various internal tick configurations. Early in training, all CTMs accurately predict parity only for initial sequence elements, gradually improving for later elements as training progresses. Models with more internal ticks achieve higher accuracy, with the 10-step model (a) correctly predicting approximately half the sequence and the 75-step model (c) correctly predicting the entire cumulative parity sequence.

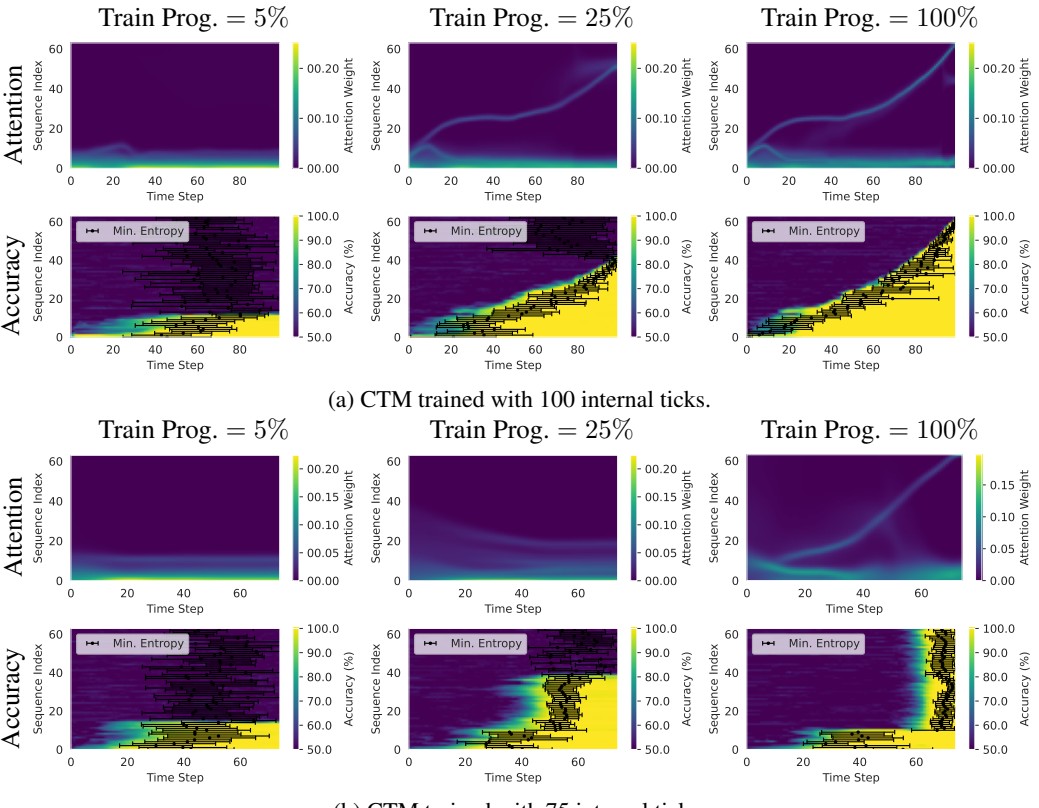

(a) CTM trained with 100 internal ticks.

(b) CTM trained with 75 internal ticks.

Figure 17: Attention patterns (top) and accuracy (bottom) at different points in training, for a CTM trained with 100 internal ticks (a) and 75 internal ticks (b). The black points in the accuracy plots denote the internal tick at which the model reached maximum certainty, with the error bars denoting one standard deviation across samples.

To gain insight into how the model solves the cumulative parity task, we visualize the CTM's attention patterns, accuracy, and points of highest certainty across all 64 elements at multiple stages of training in Figure 17 for two different models. The attention and certainty patterns evidence that these CTMs are leveraging different algorithms to solve the cumulative parity task. When using 100 internal ticks, attention moves from the beginning to the end of the sequence, and with it, the model increases its certainty of the prediction at that position. The CTM with 75 iterations, on the other hand, learns to attend to the sequence in reverse order, accurately predicting the parity of the majority of the sequence simultaneously during the final internal ticks. This reverse search through the data suggests that the CTM is carrying out a form of planning, building up its understanding of the observed data before making a final decision on the cumulative parity of the sequence. These results highlight that although multiple strategies exist for solving this task, some of which are more interpretable than others, the CTM clearly demonstrates the ability to form and follow a strategy.

### F.3 Dataset details

The input data for the parity task is a vector of length 64, where at each position is either a $-1$ or 1. The target for each sample is a vector of the same size, where at each position is the parity of the sequence up to that position, which we refer to as the cumulative parity. This data is generated on the fly, each time a new batch is fetched.

### F.4 Architecture details

The following architecture is used for the experiments in the parity task. Inputs to the model are of shape $(B, 64)$, where the size of the minibatch and sequence length are $B$ and 64, respectively. Each of the values in the 64-length sequence are either -1 or 1, at random. First, the values of -1 and 1 are converted into embeddings in $d_{embed} = d_{input}$ and positional embeddings are added. The resulting embeddings are passed through a linear layer with layer normalization to form (identical) attention keys and values. As described in section 3, the synchronization between $J_{action}$ neurons is computed and from this representation an attention query is formed. This query is then used to compute the attention values, which are concatenated to the activated state to be processed by the synapses and the neuron-level models. For the synapses we use a shallow feedforward network. This process repeats for $T$ internal ticks, where at each internal tick $t$, the synchronization can be computed between $J_{out}$ neurons and projected to the logit space.

In the parity task, we experimented with how the model performs with a varying number of internal ticks and memory length. As a baseline, we use single-layer LSTMs which are both parameter matched and use the same number of internal ticks as the CTM.

All CTM models share a common set of architectural hyperparameters, listed below. The Table 3 shows the subset of hyperparameters that vary across experimental configurations.

- $d_{model} = 1024$
- $d_{input} = 512$
- $d_{hidden} = 4$
- $k = 1$
- $p_{dropout} = 0$
- $n_{heads} = 8$
- $J_{action} = 32$
- $J_{out} = 32$
- Semi-dense pairing was used for selecting neurons for synchronization
- Absolute positional encoding was added to the input features

### F.5 Optimization details

The CTM was trained using the certainty-based loss function described in section 3.5, whereas the LSTM baselines utilized the cross-entropy loss computed at the final internal tick. This choice was

| Model | $T$ | $M$ | $d_{\text{model}}$ | Total Parameters |
|---|---|---|---|---|
| CTM | 1 | 1 | 1024 | 4908706 |
| LSTM | 1 | – | 669 | 4912710 |
| CTM | 10 | 5 | 1024 | 5043874 |
| LSTM | 10 | – | 686 | 5050716 |
| CTM | 25 | 10 | 1024 | 5212834 |
| LSTM | 25 | – | 706 | 5224386 |
| CTM | 50 | 25 | 1024 | 5719714 |
| LSTM | 50 | – | 765 | 5722374 |
| CTM | 75 | 25 | 1024 | 5719714 |
| LSTM | 75 | – | 765 | 5722374 |
| CTM | 100 | 50 | 1024 | 6564514 |
| LSTM | 100 | – | 857 | 6567486 |

Table 3: Model hyperparameters for the parity task (that vary across configurations).

made due to initial difficulties in training the LSTM effectively with the certainty-based loss. In Figure 18, we compare training accuracy curves for the LSTM baselines with 10 and 25 iterations, trained with either the final or certainty-based loss. Generally, both loss functions lead to unstable training for LSTMs with multiple internal ticks.

We used the following settings for optimization:

- Trained using a batch size of 64 on 1 H100 Nvidia GPU.

- 200000 iterations for training using AdamW [47].

- A learning rate of 1e-4 with a linear warmup of 500 iterations and decaying to zero using a cosine annealing learning rate scheduler.

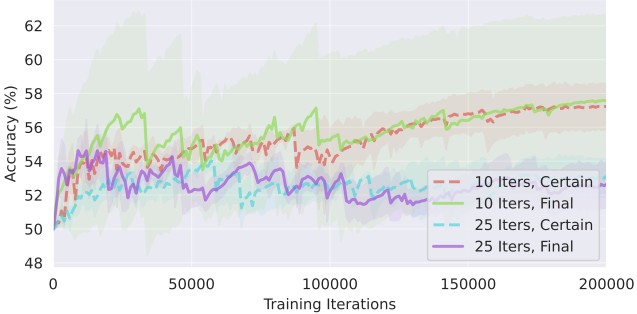

Figure 18: Test accuracies for LSTM baselines, trained with either the certainty-based loss or the cross-entropy loss at the final internal tick. Both loss functions lead to unstable learning.

## F.6 Results

**Model performance varies significantly between seeds**  Figure 6b shows the accuracy over training for various CTM and LSTM configurations, with each configuration averaged over three independent runs. These training curves exhibit considerable variance due to significant differences in performance between runs, strongly influenced by the initial random seed. For example, Figure 19 shows individual training curves for the CTM trained with 75 internal ticks and a memory length of 25. Runs 1 and 3 reach perfect accuracy, while run 2 converges to a suboptimal solution.

Furthermore, all three of these models display significantly different behaviors, with each CTM attending to very different parts of the input sequence over the 75 internal ticks. These attention patterns over the internal ticks are shown in Figure 20. Run 3 results in a model that attends from the beginning to the end of the entire sequence, while run 1 attends in reverse order.

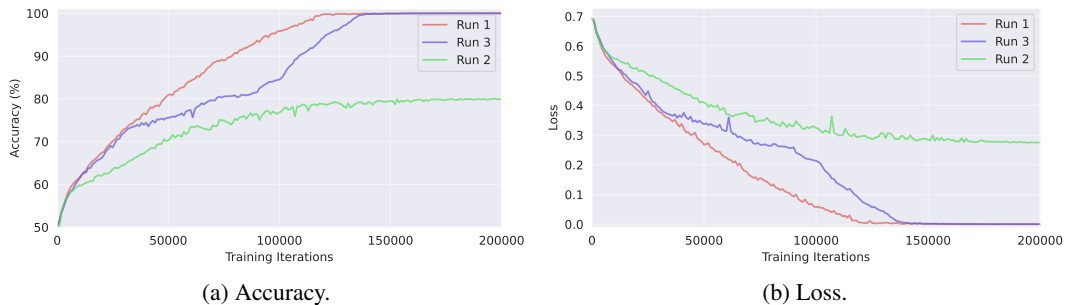

| (a) Accuracy. | (b) Loss. |

Figure 19: Training curves for three CTMs trained with three random seeds. Run 1 and 3 converge to a loss of zero, while the other run converges to a non-zero loss.



| (a) Run 1. | (b) Run 2. | (c) Run 3. |

Figure 20: Attention patterns for each of the three runs after training. For videos of these patterns emerging during training for (a) and (c) see supplementary materials '*parity-attenion-backwards.mp4*' and '*parity-attention-forward.mp4*' respectively.

## G    Additional Experiments

### G.1    CIFAR-10: CTM versus Humans and Baselines

In this section we test the CTM using CIFAR-10, comparing it to human performance, a feed-forward (FF) baseline, and an LSTM baseline. For the model-based baselines, we used a constrained featurization backbone in order to emphasize the differences owing to the model structure post-featurization (i.e., CTM versus LSTM versus FF). We also used 50 internal ticks to give the CTM and LSTM 'time to think'. The human and model baselines were set up as follows:

- **Human baseline**. We used two datasets of human labels for CIFAR-10; we call these CIFAR-10D [53] owing to its calibration of difficulty levels, and CIFAR-10H [54] originally used to quantify human uncertainty.[4] We used CIFAR-10D to determine easy versus difficult samples, and CIFAR-10H as a direct human baseline.

- **FF baseline**. A feed-forward only baseline (denoted FF). An MLP was applied to ResNet features after average pooling, where the width of the hidden layer was set to match the parameter count of the CTM for this experiment.

- **LSTM baseline**. An LSTM set up to unroll with an internal thought dimension, with a hidden width set to match the parameter count of the CTM. The LSTM could attend to the image at each step and used the same loss as the CTM for valid comparison.

Figure 21 shows the training curves of the CTM, FF, and LSTM models, and calibration plots for each, including an estimation of human calibration using CIFAR-10H. The FF baseline reaches a high training accuracy early on, but also demonstrates a poor generalization gap. The LSTM is less stable during training (we had to set the learning rate to $0.0001$ for all experiments because of this) and yields a marginally improved test accuracy. The CTM is more stable and performant.

For the human calibration we used the probabilities provided in CIFAR-10H, which were computed using guesses from multiple humans. We computed calibration here as we did for ImageNet-1K (see Figure 5b in the main paper): we compute the predictive probability as the average probability for

---

[4]CIFAR-10D can be found at `https://sites.google.com/site/hophuoctien/projects/virec/cifar10-classification`; CIFAR-10H can be found at `https://github.com/jcpeterson/cifar-10h`

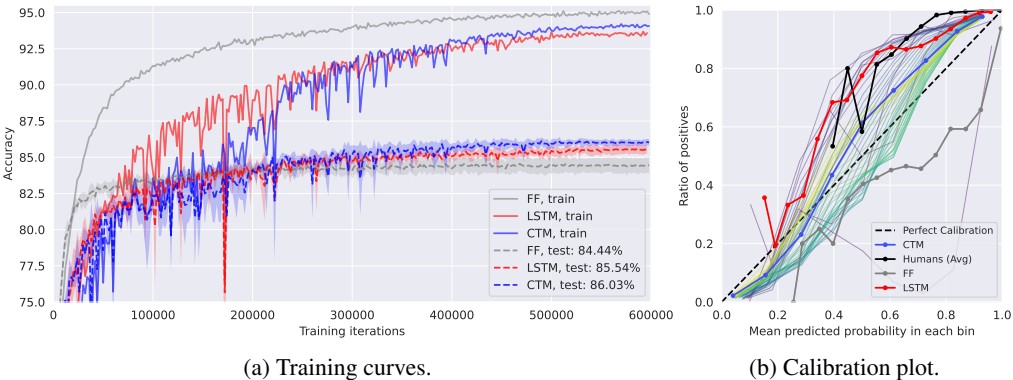

(a) Training curves.

(b) Calibration plot.

Figure 21: CIFAR-10 training curves (average over 3 seeds) and calibration plots for the CTM, a feed-forward only baseline, and an LSTM baseline. The CTM is slower than the LSTM per forward pass ($\pm 2.4 \times$) but is also more stable during learning. The CTM has the best test performance. The calibration plot shows that even a human baseline [54] is poorly calibrated, and that the CTM demonstrates good calibration, failing in a way that is strikingly similar to humans.

the chosen class over all internal ticks. None of the models are perfectly calibrated, but the CTM demonstrates the best calibration, even when compared to humans. Strikingly, the CTM has even better calibration than humans, while the LSTM follows the human under-confidence.

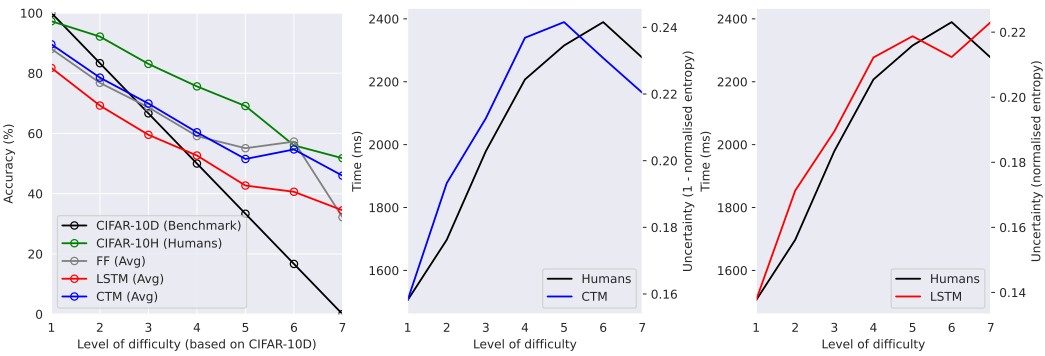

(a) Performance vs. difficulty.  (b) CTM uncertainty vs. difficulty.  (c) LSTM uncertainty vs. difficulty.

Figure 22: Analysis of model and human performance versus difficulty on CIFAR-10. We used the difficulty calibration from [53] and compared human predictions from CIFAR-10H [54]. We assume that human reaction times are a reasonable proxy for uncertainty and compare this to the trend in uncertainty for the CTM and a parameter-matched LSTM baseline. The error visualized here is a scaled standard deviation.

Figure 22a compares models and CIFAR-10H against the difficulty determined using the CIFAR-10D dataset. Each model and humans have similar trends in this case, although the CTM follows most closely to CIFAR-10H. Figures 22b and 22c compare the uncertainties of the CTM and LSTM to the uncertainties of humans (using reaction times from CIFAR-10H as a proxy for uncertainty). We compute the CTM and LSTM uncertainties using the normalized entropies (see Section 3.5 in the main paper) averaged over internal ticks as this approximates the total uncertainty each model has regarding the observed data. Both the CTM and LSTM exhibit trends similar to human reaction times.

Figure 23 shows the neural activities for the CTM and the LSTM baseline. The CTM yields rich, diverse, and complex dynamics with multiple interesting features, including periodic behavior (there is *no periodic driving function*). The distinct difference between the CTM and LSTM neural activities is evidence that the two novel elements of the CTM (NLMs and synchronization as a representation) enable neural dynamics as a fundamental computational tool.

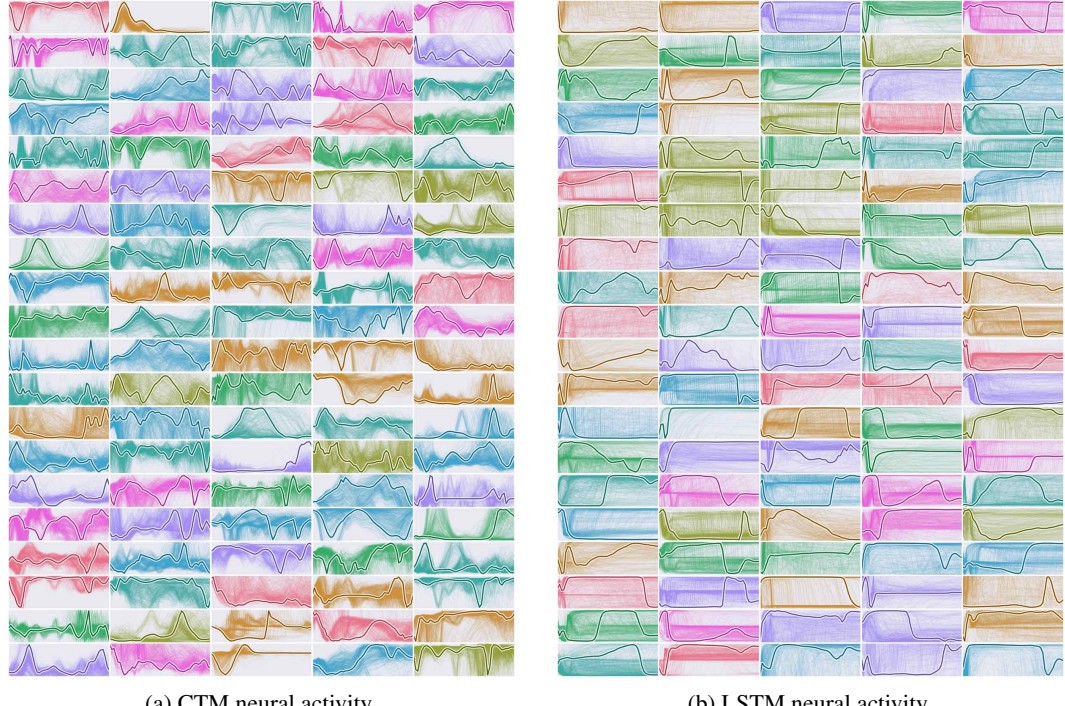

(a) CTM neural activity.         (b) LSTM neural activity.

Figure 23: Neuron traces for the CTM and an LSTM baseline on CIFAR-10, showing how the CTM produces and uses complex neural dynamics. The LSTM yields some dynamic behavior in the post-activation histories shown here, but not nearly to the same degree. Each subplot (in a random color) shows the activity of a single neuron over internal ticks, where multiple examples for different images are shown as faint background lines, and the foreground line is from a randomly chosen example.

### G.1.1 Implementation details

For this experiment, we used the first hyper-block of a constrained ResNet-18 backbone (see Appendix E.1): 5 convolutional layers in total and a downsample factor of $2\times$. We used the following hyperparameters for the CTM:

- $D = 256$ (the width of $\mathbf{z}^t$ and $\mathbf{a}^t$)
- $k = 10$ (synapse depth, 5 layers down and 5 layers up)
- $d_{\text{input}} = 64$ (the width of attention output, $\mathbf{o}^t$)
- $n_{\text{heads}} = 16$
- Random pairing for neuron selection (see Appendix C.2)
- $D_{\text{out}} = 256$ (width of $\mathbf{S}_{\text{out}}^t$ synchronization representation)
- $D_{\text{action}} = 512$ (width of $\mathbf{S}_{\text{action}}^t$ synchronization representation)
- $n_{\text{self}} = 0$
- $T = 50$
- $M = 15$ (FIFO rolling memory input to NLMs)
- $d_{\text{hidden}} = 64$ (width of MLPs inside NLMs)
- $p_{\text{dropout}} = 0.0$
- Weight decay of 0.0001
- No positional embedding

We used the following settings for optimization:

- Trained using a batch size of 512 on 1 H100 Nvidia GPU
- 600000 iterations for training using AdamW [47]
- A learning rate of 1e-4 with a linear warmup of 2000 iterations and decaying to zero using a cosine annealing learning rate scheduler

For the LSTM baseline a 2-layer LSTM was used as this performed better than a single layer LSTM setup and was relatively stable in training (compared to the maze task). The CTM synapse depth, memory length, and NLM hidden width were chosen such that the model width was kept constant (256) while parameter counts were closely matched between the CTM and LSTM. For the feed-forward model we kept the model width constant.

## G.2 CIFAR-100: Ablation Analysis

In this section we explore two aspects of the CTM: (1) width (i.e., number of neurons), and (2) number of internal ticks. We used CIFAR-100 in the experiments discussed below as it is a more challenging dataset than CIFAR-10, while remaining relatively low-demand regarding compute.

### G.2.1 Varying the number of neurons

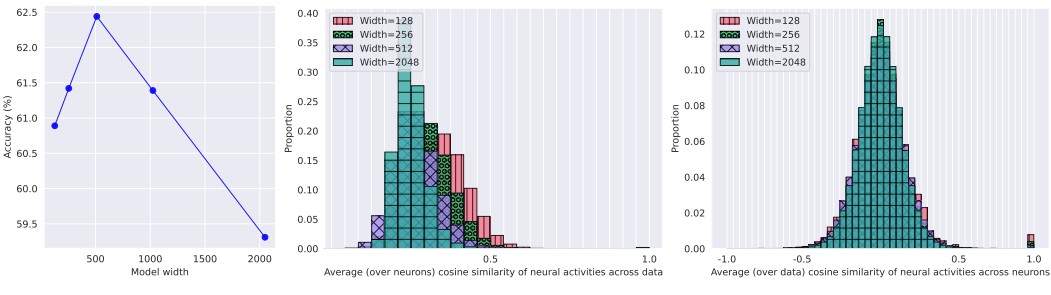

(a) CIFAR-100 accuracies.  (b) Neuron similarity across data.  (c) Neuron similarity across neurons.

Figure 24: CIFAR-100 accuracies and neuron similarities for different model widths. For (b) neuron similarity across data, we computed the average (over neurons) cosine similarities between matched neurons for all pairings across a sample of 128 images – each bar is the proportion of image pairings that have this average neuron similarity. For (c) neuron similarity across neurons, we compute the average (over data) cosine similarities for all pairs of neurons within each model – each bar is the proportion of neurons having that average cosine similarity. Cosine similarity absolutely closer to zero indicates dissimilarity, and hence improved neuron diversity.

Figure 24a shows CIFAR-100 accuracy versus model width (i.e., the number of neurons) for a fixed backbone network (details of which in Appendix G.2.3), evidencing improved test performance to a point, and then a reduction in performance. The performance drop-off might be related to overfitting, but it might also be that a wider model requires more training (we set a fixed number of training iterations).

Figures 24b and 24c show a relationship between model width and the diversity of neural activity. Intuitively, we expect that with more neurons we would observe a greater degree of neural activity, and these distributions show exactly that. In Figure 24b we see that when measuring cosine similarity on a neuron-level across data points (averaged over all neurons), a wider model results in a tighter distribution around zero. This means that a wider model results in less similar neurons, indicating that the CTM can encode more information about a data point in its neural dynamics when there are more neurons to work with. Figure 24c shows a similar quantity, where we measure the cosine similarity across neurons for the same data points (averaged over many different data points). In this case the wider model only results in a slightly tighter distribution.

### G.2.2 The impact of longer thinking

Figure 25 explores the impact of internal ticks on the CTM, showing (a) accuracies versus internal ticks and (b) the distributions over internal ticks where the CTM is most certain. The accuracies

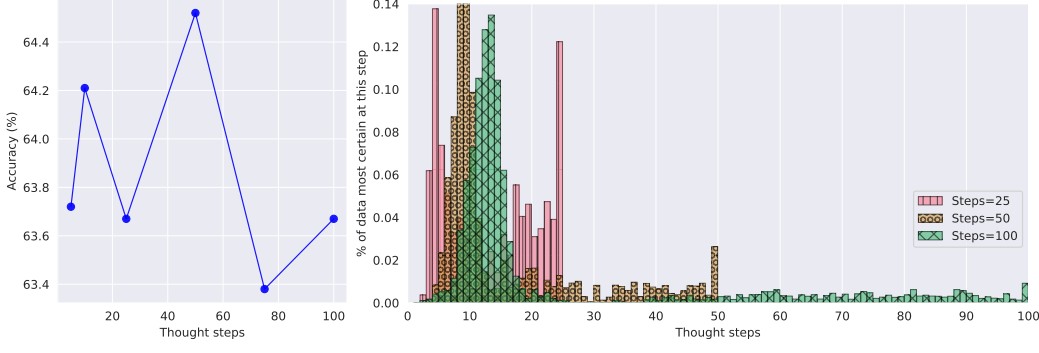

(a) CIFAR-100 accuracies.      (b) Distribution of where the CTM is most certain.

Figure 25: CIFAR-100 accuracies and internal tick analysis. The distributions and accuracies in (b) are computed for those internal ticks (x-axis) where the CTM's were the most certain (see Section 3.5 in main paper). In each case the CTM has two regions of certainty, early on and later, regardless of how many internal ticks are used.

in Figure 25a are close, although the CTM using 50 internal ticks was the most performant. This suggests once more that with more internal ticks more training might be warranted.

The emergence of two regions of high certainty in Figure 25b is interesting as it indicates that these CTMs do indeed benefit from having more 'time to think', perhaps following two different processes internally depending on the data. Although it is difficult to say exactly why this emerges, the fact that these distributions are far from uniform indicates a more complex process than simply computing the result in a strictly feed-forward nature; more analysis is required in future work.

### G.2.3 Implementation details

For Appendix G.2.1 we used a the first two hyper-blocks of a constrained ResNet-34 backbone (see Appendix E.1): a downsample factor of $4\times$. For this experiment we varied $D$ but kept all other hyperparameters as:

- $k = 8$ (synapse depth, $4$ layers down and $4$ layers up)
- $d_{\text{input}} = 512$ (the width of attention output, $\mathbf{o}^t$)
- $n_{\text{heads}} = 8$
- Random pairing for neuron selection (see Appendix C.2)
- $D_{\text{out}} = 2048$ (width of $\mathbf{S}^t_{\text{out}}$ synchronization representation)
- $D_{\text{action}} = 1024$ (width of $\mathbf{S}^t_{\text{action}}$ synchronization representation)
- $n_{\text{self}} = 32$
- $T = 50$
- $M = 25$ (FIFO rolling memory input to NLMs)
- $d_{\text{hidden}} = 32$ (width of MLPs inside NLMs)
- $p_{\text{dropout}} = 0.2$
- No weight decay
- No positional embedding

For Appendix G.2.2 we used the first two hyper-blocks of a constrained ResNet-19 backbone (see Appendix E.1): a downsample factor of $4\times$. For this experiment we varied $T$ but kept all other hyperparameters as:

- $D = 512$
- $k = 4$ (synapse depth, $2$ layers down and $2$ layers up)
- $d_{\text{input}} = 256$ (the width of attention output, $\mathbf{o}^t$)

- $n_{\text{heads}} = 4$
- Random pairing for neuron selection (see Appendix C.2)
- $D_{\text{out}} = 256$ (width of $\mathbf{S}_{\text{out}}^t$ synchronization representation)
- $D_{\text{action}} = 256$ (width of $\mathbf{S}_{\text{action}}^t$ synchronization representation)
- $n_{\text{self}} = 0$
- $M = 25$ (FIFO rolling memory input to NLMs)
- $d_{\text{hidden}} = 16$ (width of MLPs inside NLMs)
- $p_{\text{dropout}} = 0.0$
- Weight decay of 0.001
- No positional embedding

This model was set up to be more constrained compared to the other CIFAR-100 ablation because of the overhead induced by using more ticks. We explained in Appendix G.2.2 that the variants trained with longer ticks could benefit from more training, and this disparity would be greater should we use bigger models for this experiment.

## G.3  Neuron-Level Models and Synchronization: Ablation Analysis

Core to the design of the CTM are the NLMs and the use of synchronization as a representation. To understand how both of these components contribute to the performance of the CTM, we carried out ablations using a variant of the maze task of Section 4, with $15 \times 15$ sized mazes. Our ablations include four model configurations:

1. The standard CTM as detailed in Section 3.
2. A CTM without NLMs, where additional synapse model layers are introduced to maintain an equivalent number of parameters.
3. A CTM without synchronization, in which attention queries and output projections are formed directly from the post-activations $z^t$.
4. An LSTM with synchronization, where the synchronization is computed using the time series of each dimension of the LSTM's hidden state.

Each configuration was parameter matched to approximately 9M parameters and trained for $100000$ iterations. A full list of hyperparameters is shown in Table 4.

The training curves for each of the four variants are shown in Figure 26, with the final results summarized in Table 5. We find that the two augmented variants of the CTM and the LSTM with synchronization were unable to achieve a solve rate above $50\%$. The standard CTM, on the other hand, achieves a solve rate of $66\%$ and a per-step accuracy of $95\%$. These findings suggest that the combination of neuron-level models and synchronization as a representation is key to the success of the CTM.

| Model | Num Params | $d_{\text{model}}$ | $d_{\text{input}}$ | k | $D_{\text{out}}$ | $D_{\text{action}}$ | Iterations |
|---|---|---|---|---|---|---|---|
| CTM | 8967160 | 1024 | 256 | 8 | 1024 | 1024 | 50 |
| CTM (No NLMs) | 8948885 | 1045 | 256 | 10 | 1024 | 1024 | 50 |
| CTM (No Synch) | 8965112 | 1024 | 256 | 8 | - | - | 50 |
| LSTM + Synch | 8967454 | 775 | 256 | 8 | 1024 | 1024 | 50 |

Table 4: Model hyperparameters for the maze ablation experiments.

## G.4  Sorting Real Numbers

In this section, we apply the CTM to the task of sorting 30 numbers drawn from the normal distribution. Sorting real numbers was a task explored by [18] when designing RNNs for adaptive compute, and it provides a test bed for understanding the role of compute for an adaptive-compute system, such as

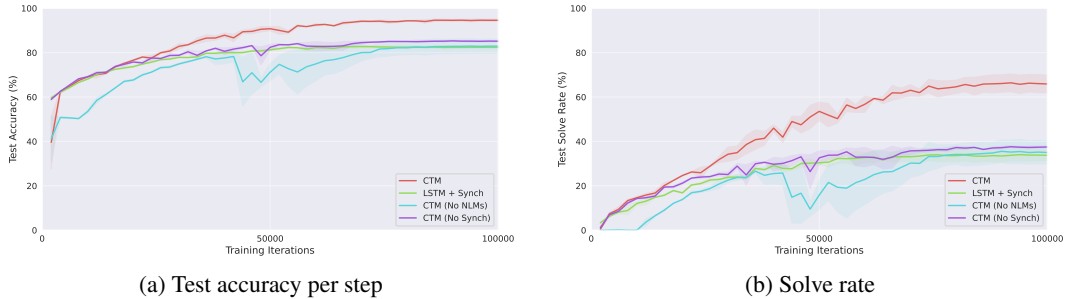

(a) Test accuracy per step

(b) Solve rate

Figure 26: Ablation analysis of NLMs and synchronization. (a) The test accuracy across training steps. (b) The solve rate across training steps. The shaded areas represent one standard deviation across two seeds.

| Method | Test Accuracy (%) | Test Solve Rate (%) |
|---|---|---|
| CTM | $94.6 \pm 0.7$ | $65.9 \pm 5.7$ |
| CTM (No NLMs) | $82.9 \pm 4.4$ | $35.0 \pm 7.2$ |
| CTM (No Synch) | $85.1 \pm 0.5$ | $37.5 \pm 0.7$ |
| LSTM + Synch | $82.4 \pm 0.9$ | $33.8 \pm 3.3$ |

Table 5: Ablation results for NLMs and synchronization.

the CTM. In this case the CTM does not use attention, but rather ingests the randomly shuffled input data (30 real numbers) directly. This is implemented by replacing the attention mechanism with a straightforward concatenation, replacing ⑩ in Figure 3.

**Can the CTM produce sequential outputs?** For this experiment we set the CTM up to output a sequence over its internal ticks. This is a more standard approach to modeling sequences and we wanted to understand whether the CTM could be trained in this fashion. At each internal tick the CTM output a vector of length 31, including 30 indices for sorting and the 'blank' token used for the well-known connectionist temporal classification (CTC) loss [46]. We then applied this CTC loss over the full output of the CTM over its internal ticks.

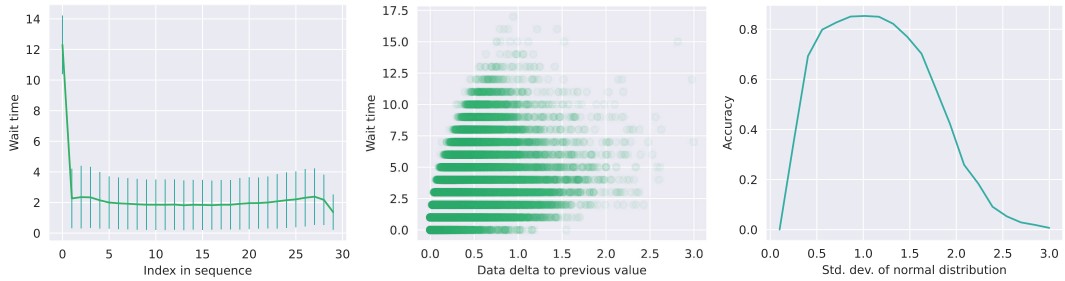

(a) Mean wait times per sequence index (measured in ticks).

(b) Wait times versus change in sequence values.

(c) Generalizing beyond the distribution used for training.

Figure 27: Results when sorting on $\mathcal{N}(0, I_{30})$. In (a) we can see an evident pattern in the average wait times, where the initial wait time (number of internal ticks) is high, goes to its lowest point, and has a slightly higher bump toward the end of the sequence. In (b) we see that the CTM employs various wait times, but that the difference between the previous output value and the current output value ('data delta') impacts wait time. In (c) we see how this CTM can scale to data drawn from different normal distributions.

Figure 27 gives the results of the CTM on the sorting task. There is a clear pattern to the process it follows, as evidenced by a correlation between wait times and both the current sequence index (a) and the difference between the previous value and the current value being output (b). A similar

task was explored by [18], who sorted 15 numbers using an adaptive compute RNN. In their case, they observed similar wait times before beginning output (analogous to our first sequence element) and also near the end of the sequence. Our analysis of the relationship between wait times and the difference between current and previous data values (what we call 'data delta' in Figure 27b) constitutes evidence that the CTM is using an internal algorithm that depends on the layout of the data. We also show that this CTM generalizes to distributions outside of the training data.

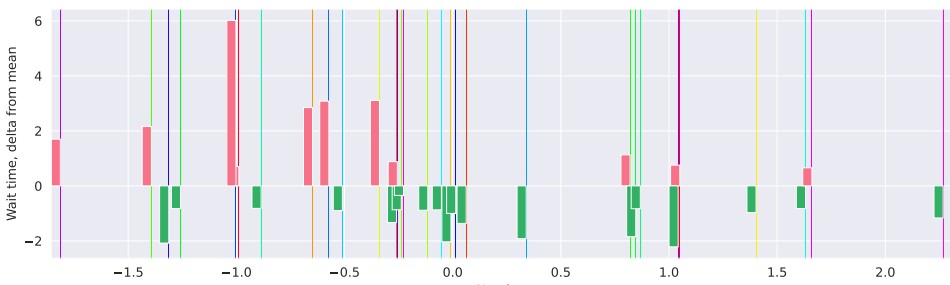

Figure 28: Sorting demonstration. The input data is represented as vertical lines whose colors denote their original shuffled position (from purple through to red in the 'rainbow' colormap). The red and green bars show positive and negative deviation from the mean wait time (Figure 27a for each index in the sequence), respectively.

Figure 28 demonstrates the CTM's wait times in a real use-case. The red bars indicate longer than average wait times for a given index, and green bars indicate shorter than average wait times. Longer wait times tend to be related to bigger gaps between data points ('data delta' in Figure 27b).

### G.5 Q&A MNIST: Memory and Arithmetic

To assess the CTM's capabilities for memory, retrieval, and arithmetic computation, we devise a Question and Answering (Q&A) MNIST task, reminiscent of [55] or [56]. In this task, the model sequentially observes a series of MNIST digits [57], followed by an interwoven series of index and operator embeddings that determine which of the observed digits to select and which modular operation to perform over them. This allows us to probe whether the CTM can simultaneously recognize hand-drawn digits, recall previous observations, and perform logical computation on them without any prior knowledge of the digits depicted in the images or the relationships between them. Furthermore, by applying more operations at inference time than observed during training time, we can test the generalizability of the CTM.

Specifically, the model first observes $N_d$ MNIST digits sequentially for $t_d$ internal ticks each. Next, the model receives an interwoven sequence of $N_{idx}$ index embeddings (indicating which digit to select) and $N_{op}$ operator embeddings (specifying either modular addition or subtraction, where each intermediate result is taken modulo 10 to keep answers within the range 0–9), each presented for $t_{idx}$ and $t_{op}$ internal ticks, respectively. Finally, the model observes a zero tensor for $t_{ans}$ internal ticks, signaling the model to produce its answer. The target, between 0 and 9, results from the composition of all specified modular arithmetic operations. An example is shown in Figure 29.

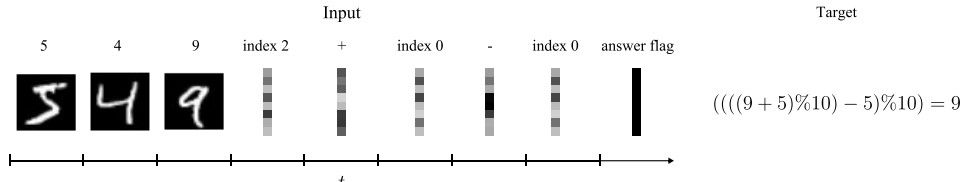

Figure 29: Overview of the Q&A MNIST task. The model observes a series of digits followed by a series of index and operator embeddings, each repeated for several internal ticks. The model is then shown an answer flag and must predict the result of the modular operations.

We trained CTMs and parameter-matched LSTMs with two different configurations, varying how many internal ticks were used to process each input. Digits and embeddings were observed for either

1 or 10 internal ticks, with corresponding answering times of 1 or 10 internal ticks. The number of digits and the number of operations were sampled uniformly between 1 and 4. Memory lengths for the 1 and 10 internal ticks per input CTMs were set to 3 and 30 steps, respectively. We highlight that with these observation and memory length configurations that the digit observations will always lie outside of the memory length-sized window during the answering stage. In this way, the CTM must organize its activations such that it can recall the digits at later time steps. The CTM is trained with the loss defined in Section 3.5 (main paper), computed only over the final $t_{\text{ans}}$ steps. Once more, we set $t_2$ to be the final iteration for the LSTM for stable training.

### G.5.1 Results

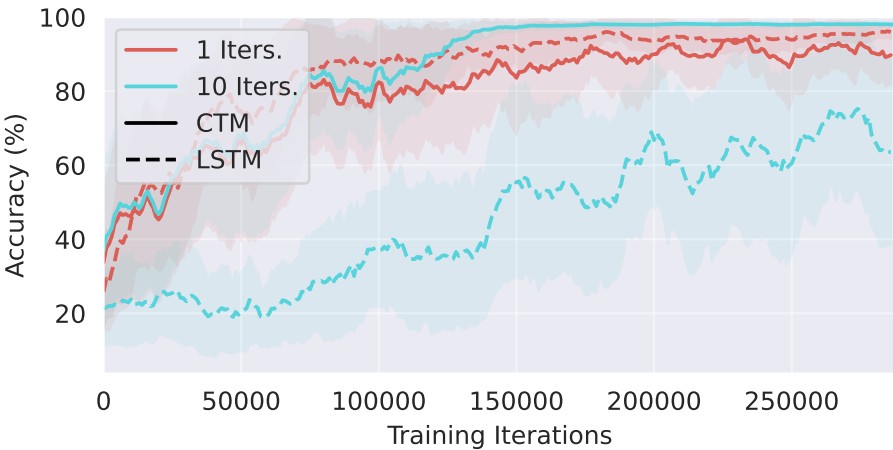

Figure 30: Training curves for the CTM and LSTM on the Q&A MNIST task. The shaded areas represent one standard deviation across seeds. With a single internal tick, the LSTM outperforms the CTM. However, the performance of the CTM increases with the number of internal ticks, while the LSTM becomes increasingly unstable.

**Memory via synchronization.**  Training curves for three seeded runs for CTMs and parameter-matched LSTMs are shown in Figure 30. With a single internal tick, the LSTM initially outperforms the CTM. As the number of internal ticks increases, the LSTM's performance degrades and learning becomes considerably more unstable. In contrast, the CTM consistently improves its performance with additional thinking time. Specifically, all three seeded runs for the CTM with 10 internal ticks per input achieved over 96% accuracy on the most challenging in-distribution task (performing four operations after observing four digits). In contrast, the corresponding 10-internal tick LSTM performed at or below 21% accuracy across all seeded runs. The strong performance of the single-tick LSTMs highlights the effectiveness of the LSTM's complex gated update, however, this mechanism does not scale effectively to multiple internal steps, unlike the CTM, which effectively utilizes internal ticks to build up a synchronization representation.

The CTM performs well even when the observed digits are outside of the memory window, indicating that it has learned to memorize what it has observed to some degree, purely via the organization and synchronization of neurons. The strong performance of the CTM indicates that processing timing information through the synchronization of neuron activations may be a powerful mechanism for memorization and recall.

**The CTM can generalize.**  We examine generalization by measuring the accuracy of the models when given more digits or index-operator embeddings than used during training. Figure 31 shows the accuracy of the CTM and LSTM as a function of the number of digits shown and operations to perform, with the training regime highlighted in red. We find that both the CTM and LSTM baselines can generalize to an increased number of operations. To understand how the model is capable of generalizing out of distribution, we illustrate an example thought process of the CTM in Figure 32, which shows a sample sequence of inputs and a snapshot of the output logits. We find that the CTM sequentially computes the modular computation as the embeddings are observed, instead of waiting for the final answer flag to determine the final solution at once. A similar behavior can be seen in the

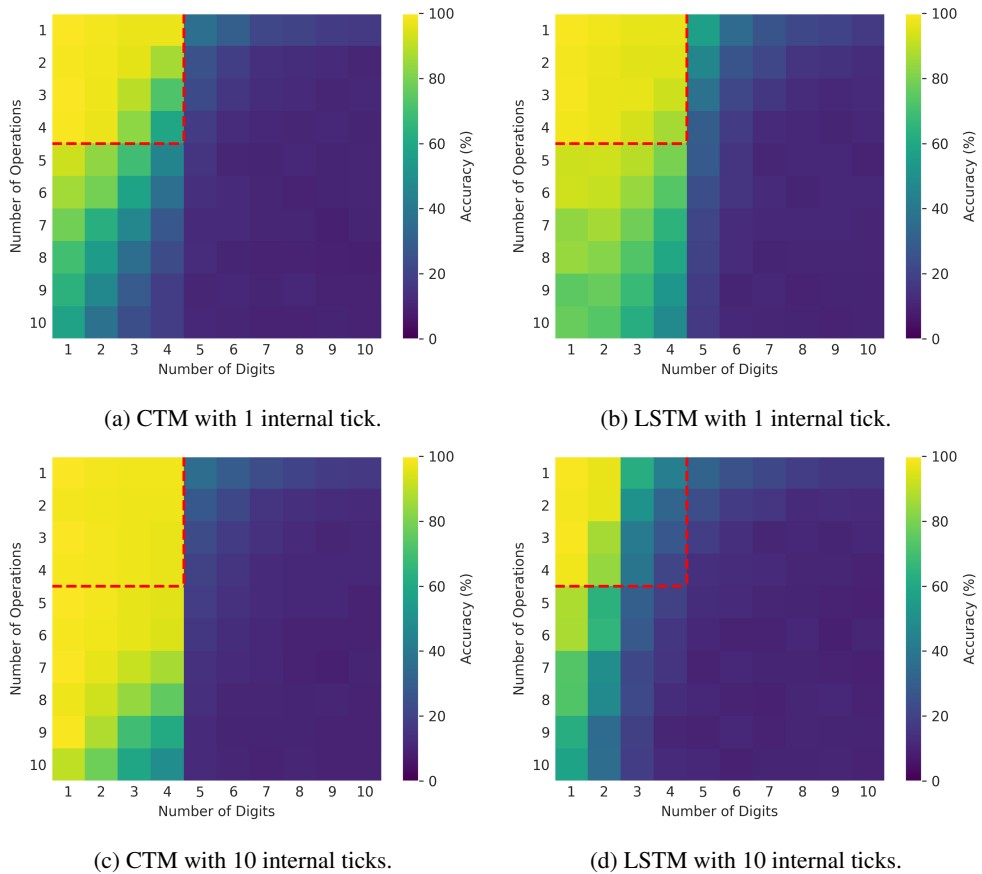

(a) CTM with 1 internal tick.

(b) LSTM with 1 internal tick.

(c) CTM with 10 internal ticks.

(d) LSTM with 10 internal ticks.

Figure 31: Generalizability on the Q&A MNIST task for CTM and LSTM models with 1 and 10 internal ticks. The x-axis denotes the number of MNIST digits input to the model, the y-axis denotes the number of operations the model must perform, and the color corresponds to the test accuracy.

1-internal tick LSTM baseline. We are not claiming that the CTM can do something that the LSTM cannot, but instead that it can learn to **use synchronization as a tool** to solve this task, and that the result is both effective and scales to longer task requirements.

### G.5.2 Implementation details

Unlike other tasks, the Q&A MNIST task processes multiple input types: MNIST digit images, embeddings for operator and index markers, and zero tensors as answer flags. MNIST images undergo preprocessing through a convolutional backbone consisting of two convolutional blocks, each containing a convolutional layer, batch normalization, a ReLU activation, and a max pooling layer. Outputs from this backbone form attention keys and values, which the CTM queries using projections from the synchronization representation. The resulting attention outputs are concatenated with the CTM's activated state before synaptic processing. In contrast, operator and index embeddings, as well as answer flags, bypass the convolutional backbone and attention mechanism, being directly concatenated to the CTM's activated state. Operators use learned embeddings, indices utilize sinusoidal embeddings [40], and answer flags are zero vectors matching the embedding dimension.

For comparison, parameter and internal tick matched single-layer LSTM baselines were used. The common parameters used in the experiment are as follows:

- $d_{model} = 1024$
- $d_{input} = 64$
- $d_{hidden} = 16$

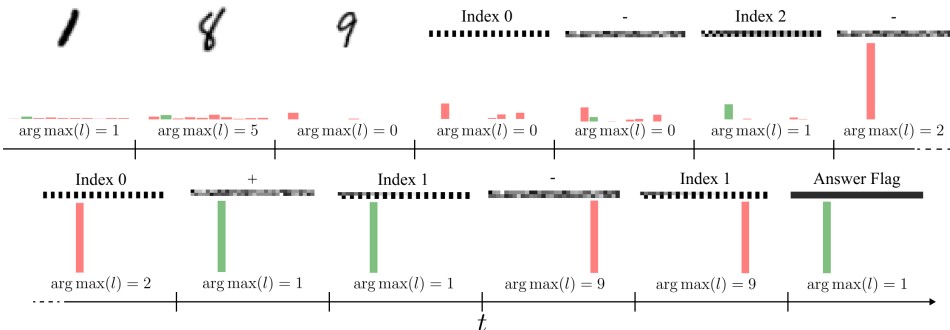

Figure 32: Example CTM thought process from the Q&A MNIST task. Shown are the inputs to the model (MNIST digit, index and operator embeddings) as well as the argmax of the output logits $l$, at different snapshots. Each input is repeated for 10 internal ticks. In this case, the model is to compute $((((((1 - 9)\%10) - 1)\%10 + 8)\%10 - 8)\%10)$. We find that the model computes each part of this composition sequentially as the embeddings are observed, with outputs of 2, 1, 9, and finally, the correct answer of 1, projected from the synchronization representation.

- $k = 1$
- $p_{\text{dropout}} = 0$
- $n_{\text{heads}} = 4$
- $J_{\text{action}} = 32$
- $J_{\text{out}} = 32$
- Semi-dense pairing was used for selecting neurons for synchronization.
- Positional encoding was not used.

Detailed hyperparameters of the CTM are provided in Table 6.

| Model | $T$ | $M$ | Repeats/Input | Answering Steps | Total Parameters |
|---|---|---|---|---|---|
| CTM | 1 | 3 | 1 | 1 | 2,501,388 |
| LSTM | 1 | – | 1 | 1 | 2,507,218 |
| CTM | 10 | 30 | 10 | 10 | 3,413,772 |
| LSTM | 10 | – | 10 | 10 | 3,418,954 |

Table 6: Differing model hyperparameters and total parameters for the Q&A MNIST experiments. The column **Repeats/Input** refers to the number of internal ticks the model used to process a unique input. For example, **Repeats/Input = 10** implies that 10 internal ticks are used to process each MNIST digit and each index or operator embedding. The **Answering Steps** refers to the number of internal ticks the answering flag is observed for.

The CTM was trained using the certainty-based loss function described in section 3.5, while the LSTM baselines were trained using the cross-entropy loss at the final internal tick. We used the following settings for optimization:

- Trained using a batch size 64 on 1 H100 Nvidia GPU.
- 300000 iterations for training using AdamW [47].
- A learning rate of 1e-4 with a linear warmup of 500 iterations and decaying to zero using a cosine annealing learning rate scheduler.

## G.6 Reinforcement Learning

We have previously shown that the CTM can process sequentially on non-sequential tasks via its decoupled internal recurrence. Here, we extend the CTM to sequential decision-making tasks involving interactions with external environments. Specifically, we train CTMs using reinforcement

learning (RL), where the model learns action-selection policies based on environmental observations and trial-and-error interactions. In this setting, the CTM processes one or more internal ticks before producing an action that transitions the environment to the next state. To achieve this, we continuously maintain the neuron dynamics across these internal ticks over successive environment steps, allowing previous environmental observations to influence current internal states via the NLMs. A central goal of this section is to provide evidence that the CTM can be set up to learn in a continuous environment.

### G.6.1 Implementation Details

**Environments.** We test the CTM on two classic control tasks and one navigation task, namely, CartPole, Acrobot and MiniGrid Four Rooms, implemented in Gymnasium [58, 59, 60, 61]. Examples of these tasks are shown in Figure 33. Because the CTM maintains an activation history across environment transitions, it functions as a stateful recurrent neural network. Therefore, we specifically evaluate the CTM in partially observable settings, where RNNs are effective [62]. Partial observability is introduced by masking the positional and angular velocity observation components in the control tasks and restricting the field of view in the navigation task. This masking converts these tasks into partially observable Markov decision processes (POMDPs), requiring the CTMs to develop policies that recall past observations. For instance, in the Acrobot task, selecting the correct action depends on recalling past positions and inferring the velocity to increase arm elevation.

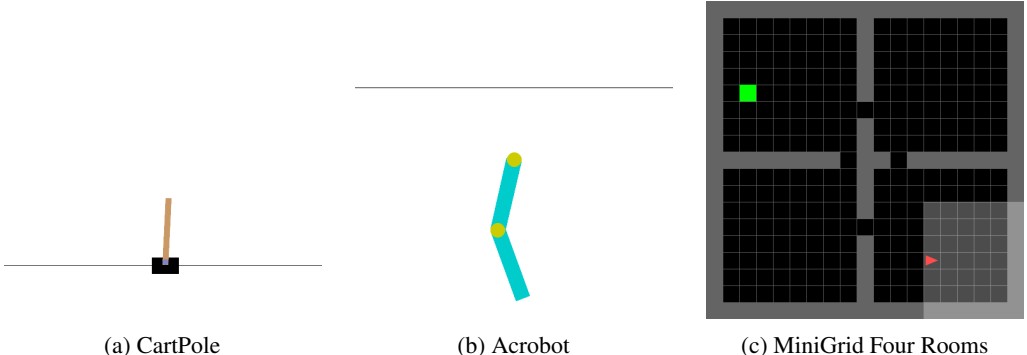

(a) CartPole        (b) Acrobot        (c) MiniGrid Four Rooms

Figure 33: Reinforcement learning environments used for CTM evaluation.

**Architecture.** The configuration of the CTM for training with PPO is as follows. First, observations are processed by a backbone, such as a feedforward network, and are concatenated to the current activated state of the CTM, without an attention mechanism, for processing over a fixed number of internal ticks. After this fixed number of internal ticks, the synchronization between the output neurons is calculated and passed to the actor and critic heads for selecting the next action and estimating the state value.

Unlike the other tasks, which calculate synchronization across the entire activation history, in the RL setting we use a sliding window of size memory length $M$. This approach prevents the buildup of very long activation histories, which may grow into the thousands for these tasks. Additionally, this allows for maintaining tensors of the same shape across all stages of the episode rollouts. To facilitate this, the CTM is initialized with both a learned initial state trace and a learned initial activated state trace, which are supplied to the model on the initialization of each episode. After a single forward pass of the model (corresponding to a single environment step), these state traces will be maintained and provided to the model on the next environment step. This allows the CTM to process a continuous history of activity, enabling activations from many environment states in the past to impact the present.

For the classic control tasks, the observation backbone is composed of two blocks containing a linear layer, a gated linear unit (GLU) [63] and layer normalization. A similar input processing is carried out for the navigation task, however, each of the object, color and state IDs are first embed in $d_{embed} = 8$. While for the CTM the output of the backbone is concatenated to the current activated state, for the LSTM baseline, the output is instead processed for the same number of internal ticks as the CTM. The actor and critic heads are implemented as two-layer multilayer perceptrons (MLPs), each comprising two hidden layers of 64 neurons with ReLU activations. For the CTM, these heads

receive the synchronization of the output neurons as inputs, while for the LSTM baselines, they receive the hidden state of the LSTM after $T$ internal tick. Dense pairing was used to select neurons for synchronization.

Unlike other tasks such as image classification (Section 5), which use a UNet-style synapse model, the RL tasks employ a two-layer feedforward synapse, where each layer consists of a linear transformation, a GLU and LayerNorm. Empirically, we found that two of these layers significantly outperformed a single layer, particularly in the navigation task, where a single-layer synapse consistently failed to match the LSTM's average episode length.

The model hyperparameters used for the experiments for CartPole, Acrobot and MiniGrid Four Rooms can be found in Tables 7 to 9. The PPO implementation is based on [64].

| Model | $T$ | $M$ | $d_{\text{model}}$ | $d_{\text{input}}$ | $d_{\text{hidden}}$ | $J_{\text{out}}$ | Total Parameters |
|---|---|---|---|---|---|---|---|
| CTM | 1 | 10 | 128 | 128 | 4 | 16 | 175437 |
| LSTM | 1 | – | 118 | 128 | – | – | 175855 |
| CTM | 2 | 20 | 128 | 128 | 4 | 16 | 188237 |
| LSTM | 2 | – | 126 | 128 | – | – | 188863 |
| CTM | 5 | 50 | 128 | 128 | 4 | 16 | 226637 |
| LSTM | 5 | – | 148 | 128 | – | – | 227275 |

Table 7: Model hyperparameters for the CartPole experiments.

| Model | $T$ | $M$ | $d_{\text{model}}$ | $d_{\text{input}}$ | $d_{\text{hidden}}$ | $J_{\text{out}}$ | Total Parameters |
|---|---|---|---|---|---|---|---|
| CTM | 1 | 5 | 256 | 64 | 4 | 16 | 350094 |
| LSTM | 1 | – | 243 | 64 | – | – | 350118 |
| CTM | 2 | 10 | 256 | 64 | 4 | 16 | 362894 |
| LSTM | 2 | – | 249 | 64 | – | – | 364290 |
| CTM | 5 | 25 | 256 | 64 | 4 | 16 | 401294 |
| LSTM | 5 | – | 265 | 64 | – | – | 403490 |

Table 8: Model hyperparameters for the Acrobot experiments.

| Model | $T$ | $M$ | $d_{\text{model}}$ | $d_{\text{input}}$ | $d_{\text{hidden}}$ | $J_{\text{out}}$ | Total Parameters |
|---|---|---|---|---|---|---|---|
| CTM | 1 | 10 | 512 | 128 | 16 | 32 | 7802690 |
| LSTM | 1 | – | 294 | 128 | – | – | 7813692 |
| CTM | 2 | 20 | 512 | 128 | 16 | 32 | 7976770 |
| LSTM | 2 | – | 300 | 128 | – | – | 7979304 |

Table 9: Model hyperparameters for the MiniGrid Four Rooms experiments.

**Opimization.** The models were trained with Proximal Policy Optimization [65] on single H100 Nvidia GPU. The same set of PPO hyperparameters were used for both the CTM and the LSTM baseline, and are shown in Table 10.

### G.6.2 Results

**The CTM can continuously interact with the world.** Training curves for the reinforcement learning tasks are shown in Figure 34. In all tasks, we find that the CTM achieves a similar performance to the LSTM baselines.

Figure 35 compares the neuron traces of the CTM and the LSTM baselines for the CartPole, Acrobot and MiniGrid Four Rooms tasks. In the classic control tasks, the activations for both the CTM and the LSTM feature oscillatory behavior, corresponding to the back-and-forth movements of the cart and arm. For the navigation task, a rich and complex activation pattern emerges in the CTM. The LSTM on the other hand, features a less diverse set of activations. The LSTMs trained in this section have a more dynamic neural activity than what can be seen when trained on CIFAR-10 (Figure 23).

| Hyperparameter | CartPole | Acrobot | MiniGrid Four Rooms |
|---|---|---|---|
| Learning Rate (LR) | $1 \times 10^{-3}$ | $5 \times 10^{-4}$ | $1 \times 10^{-4}$ |
| Total Environment Steps | 10M | 2M | 300M |
| Rollout Length | 50 | 100 | 50 |
| Number of Environments | 256 | 12 | 256 |
| Max Environment Steps per Episode | 200 | 500 | 300 |
| Update Epochs | 4 | 1 | 1 |
| Minibatches | 4 | 4 | 4 |
| Discount Factor ($\gamma$) | 0.99 | 0.99 | 0.99 |
| GAE Lambda ($\lambda$) | 0.95 | 0.95 | 0.95 |
| Clip Coefficient | 0.1 | 0.1 | 0.1 |
| Entropy Coefficient | 0.1 | 0.1 | 0.1 |
| Value Function Coefficient | 0.25 | 0.25 | 0.25 |
| Value Function Clipping | No | No | No |
| Max Gradient Norm | 0.5 | 0.5 | 0.5 |

Table 10: PPO hyperparameters for each task.

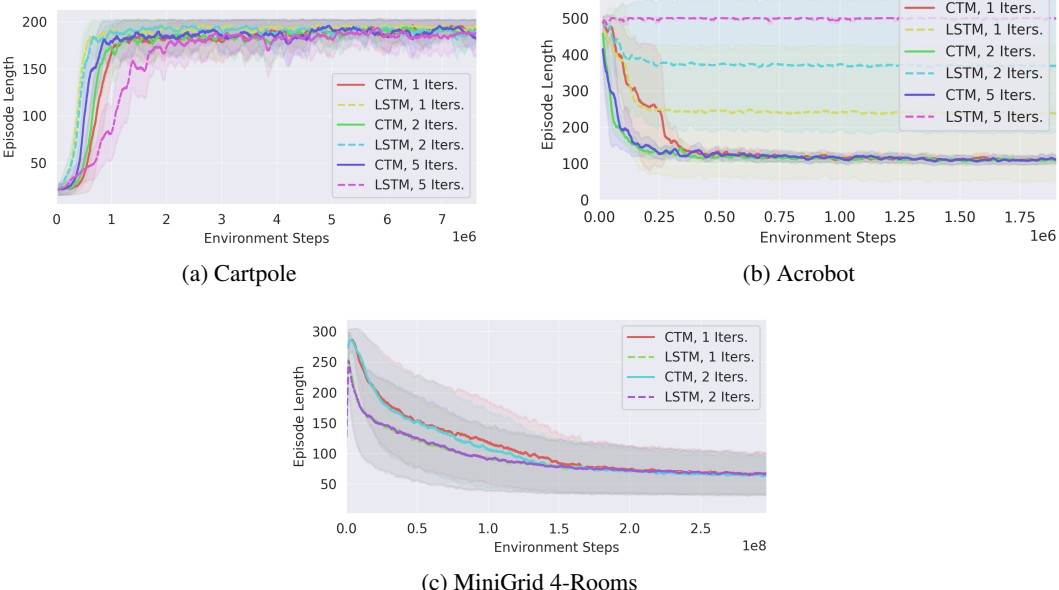

(a) Cartpole

(b) Acrobot

(c) MiniGrid 4-Rooms

Figure 34: Training curves for reinforcement learning tasks. Each curve depicts a moving average of the episode length during training, averaged over three training runs. The shaded region represents one standard deviation across seeds. For Cartpole, higher is better. For Acrobot and MiniGrid 4-rooms, lower is better.

This is likely due to the sequential nature of RL tasks, where the input to the model changes over time owing to its interaction with the environment, inducing a feedback loop that results in the model's latent representation also evolving over time.

# H  Recursive computation of the synchronization matrix

In Section 3.4 we defined the synchronization matrix at internal tick $t$ as

$$\mathbf{S}^t \;=\; \mathbf{Z}^t \, (\mathbf{Z}^t)^{\mathsf{T}}, \qquad \mathbf{Z}^t \in \mathbb{R}^{D \times t}, \tag{12}$$

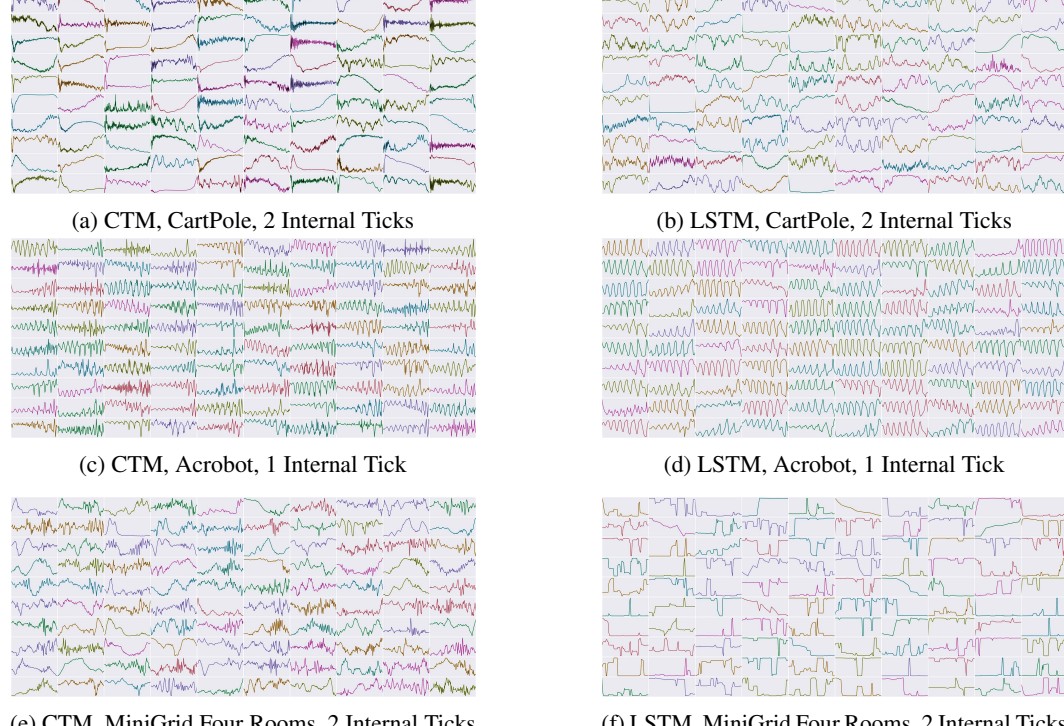

(a) CTM, CartPole, 2 Internal Ticks

(b) LSTM, CartPole, 2 Internal Ticks

(c) CTM, Acrobot, 1 Internal Tick

(d) LSTM, Acrobot, 1 Internal Tick

(e) CTM, MiniGrid Four Rooms, 2 Internal Ticks

(f) LSTM, MiniGrid Four Rooms, 2 Internal Ticks

Figure 35: Neural activities over the course of a single episode for the CTM and LSTM on CartPole, Acrobot and MiniGrid Four Rooms tasks. The CTM features richer neuron dynamics than the LSTM.

where the $d$–th row of $\mathbf{Z}^t$ stores the post–activation trace of neuron $d$ up to tick $t$ (cf. Eq. (4)). Because Eq. (12) recomputes all $D^2$ inner products from scratch at every tick, its time complexity is $\mathcal{O}(D^2 t)$ over a roll-out of length $t$. Below we show that, with the exponentially–decaying rescaling of Eq. (10), the same quantity can be obtained from a pair of first–order recursions that require only $\mathcal{O}(D_{\mathrm{sub}})$ work per tick, where $D_{\mathrm{sub}} \ll D$ is the number of subsampled neuron indices actually used for the output and action projections.

For notational clarity we first consider a single $(i, j)$ neuron pair and omit the subsampling; the extension to a batch of pairs is immediate. Recall that the rescaled synchronization entry is defined as

$$S_{ij}^t = \frac{\sum_{\tau=1}^{t} \mathrm{e}^{-r_{ij}(t-\tau)} z_i^\tau z_j^\tau}{\sqrt{\sum_{\tau=1}^{t} \mathrm{e}^{-r_{ij}(t-\tau)}}}, \tag{13}$$

where $r_{ij} \geq 0$ is the learnable decay rate for the pair $(i, j)$. Define the following auxiliary sequences

$$\alpha_{ij}^t := \sum_{\tau=1}^{t} \mathrm{e}^{-r_{ij}(t-\tau)} z_i^\tau z_j^\tau, \qquad\qquad \alpha_{ij}^1 = z_i^1 z_j^1, \tag{14}$$

$$\beta_{ij}^t := \sum_{\tau=1}^{t} \mathrm{e}^{-r_{ij}(t-\tau)}, \qquad\qquad \beta_{ij}^1 = 1. \tag{15}$$

Then $S_{ij}^t = \alpha_{ij}^t / \sqrt{\beta_{ij}^t}$ and both $\alpha_{ij}^t$ and $\beta_{ij}^t$ obey simple first–order difference equations:

$$\alpha_{ij}^{t+1} = \mathrm{e}^{-r_{ij}} \alpha_{ij}^t + z_i^{t+1} z_j^{t+1}, \tag{16}$$

$$\beta_{ij}^{t+1} = \mathrm{e}^{-r_{ij}} \beta_{ij}^t + 1. \tag{17}$$

The rank–1 update in Eq. (16) makes it unnecessary to store the full activation history or to repeatedly form large outer products. During forward simulation we maintain $\alpha_{ij}^t$ and $\beta_{ij}^t$ for each selected pair and update them in $\mathcal{O}(1)$ time.

In practice we store $\{\alpha_{ij}^t, \beta_{ij}^t\}$ only for the two disjoint subsamples that form $\mathbf{S}_{\text{out}}^t$ and $\mathbf{S}_{\text{action}}^t$ (Section 3.4). Both memory footprint and compute overhead therefore scale linearly with the number of retained pairs, i.e. $\mathcal{O}(D_{\text{sub}}) = \mathcal{O}(D_{\text{out}} + D_{\text{action}})$ per tick.

# I   Emergent phenomena

We found that there were several emergent behaviors during learning and when the CTM was put under strong constraints (e.g., limited internal ticks), and we encourage the reader to train their own CTMs (code included in supplementary material) in order to make the same observations. These emergent behaviors include:

1. **At initialization** the neural dynamics are poor and the neurons **do not exhibit any periodic nature**. The periodic nature of the dynamics (as we have shown in this paper) only emerge as training progresses, and the dynamics become richer, more complex and diverse, and periodic as training continues. One can apply layer-norm to pre-activation histories (inputs to NLMs) in order to drive more periodic behavior at the outset, but the performance is generally worse (there is an option for this in the code).

2. Regarding neural dynamics, a useful emergent property of the CTM architecture is that visualizing neural dynamics enables understanding whether **there are any 'dead' neurons** (there is actually 1 of these in Figure 2a, evidenced by having extremely little variation across data instances). This enables a new perspective regarding the utilization of weights in a NN, and we hope that future work can leverage this emergent utility.

3. For the maze task we found that relatively early on in training the CTM would **exhibit a 'double take' approach** where it would rapidly and approximately solve the maze (as shown by the attention progression) only to start again and do so more slowly. We think that this is a naturally **bootstrapping technique** as it disappears as the CTM gets better at solving the maze (it is wasted compute, after all).

4. Sometimes the CTM begins down the **incorrect path** for the maze task but soon **changes its mind** and follows the correct path. This is a consequence of the freedom afforded by our loss function. Supplementary video '*maze-change.mp4*' demonstrates an instance of this.

5. Upon careful inspection of the attention heads for the maze task we noticed that **some heads were dedicated to a broader, more global perspective** of the maze, while other heads were dedicated to following the path from start to end points. A similar behavior emerged in the parity task, where some heads would be dedicated to finding positive (or negative) values (see supplementary video '*parity.mp4*').

6. On ImageNet-1K we found that the degree to which the CTM **'looks around' also increases with training time**, becoming more diverse.

7. Also on ImageNet-1K we observe some evidence that the CTM tends to shift between **broad perspectives** (spread out attention) **to narrow views** (tight attention) as it thinks.

8. Also on ImageNet-1K, even without positional embeddings the CTM could learn to **follow directional regions** over real images in much the same way as it did for mazes (see supplementary video '*imagenet.mp4*').

9. Under heavy **constraints**, such as when using limited internal ticks (e.g., 50) and requiring long solutions (e.g., 150 or 200 steps in a maze; we showed only 100 steps in this paper), we found that the CTM would **implement unusual or alternative strategies**. These strategies would still remain interpretable but could be more computationally efficient. We include a supplementary video '*maze-backwards.mp4*' demonstrating an interesting maze solution where a constrained CTM ended up learning a highly-performant strategy that looked ahead by small chunks before then tracing **backwards**, jumping forwards again, and repeating until the maze was solved. We find this behavior fascinating and evident of what a machine could do that might be strange to humans, yet still understandable.

10. CTMs that can perfectly solve the parity task **consistently learned two types of strategies**: either attending to the data from the beginning to end, or in reverse order. We show two examples in supplementary videos '*parity-attenion-forward.mp4*' and '*parity-attention-backwards.mp4*', which highlight how these behaviors emerge during training.

11. In the Q&A MNIST task (Appendix G.5), we found that the CTM would **emit the intermediate modular result** after each index–operator–index tuple, rather than waiting for the answering flag to produce the final output. Although LSTMs can display this behavior with a single internal tick, they struggle to maintain it as the number of internal ticks grow.

