# OpenReview forum: "Continuous Thought Machines"
_NeurIPS.cc/2025/Conference — NeurIPS 2025 spotlight_

### Official Review · Reviewer_Fj9J · 2025-06-16

**Clarity:** 4
**Significance:** 3
**Originality:** 3
**Rating:** 5
**Confidence:** 4

**Summary:**

This paper proposes a new model for deep learning inspired by two important (and underexplored) ideas in neuroscience: (i) neuron-specific dynamics, and (ii) across-neuron synchronization. The resulting model is essentially a recurrent neural network, where each "neuron" is parameterized by a complex neural network, namely one feedforward network to compute its current potential as a function of its neighbors' activity, and a second recurrent network to compute its output as a function of its historical potential (this captures idea (i)). Property (ii) arises in that the model's actual output is (a projection of) the similarity matrix of the trajectories of its neurons -- in other words, their synchronization. The model is designed to adapt its computational time, particularly through a loss function that incentivizes it to simply make its prediction its most confident output, regardless of timestep. The model is evaluated through a diverse variety of tasks with both sequential and non-sequential character, including image classification and solving mazes.

**Questions:**

Q1. Of course, it would be very interesting to see a language experiment in a model like this.
Q2. Why is it necessary to include both the loss on the best tick and the loss on the most confident tick in the loss function? Intuitively, it seems like only the second term is important.
Q3. The "Emergent Phenomena" section of the appendix is interesting and it would be nice to see a few of these observations empirically fleshed out in the main body.

**Ethical Concerns:**

["NO or VERY MINOR ethics concerns only"]

**Final Justification:**

I believe this is an insightful and strong paper which will introduce a few interesting ideas to the community. I'm doubtful that any of those ideas will be transformative but they are well worth sharing. I have no concerns with the rigor with which the paper's claims were evaluated. Another reviewer raised an issue over the authors' novelty claim regarding synchronization -- however, the authors immediately recognized the issue and were willing to adjust the language. The claim was overblown but the method remains novel. In sum, this paper is a clear 5, accept for me.

**Limitations:**

Yes

**Paper Formatting Concerns:**

No issues

**Quality:**

3

**Strengths And Weaknesses:**

Strengths:
S1. The writing and presentation throughout are excellent. The model is fairly complex so this is very important. The supplementary website with animations is a nice bonus.
S2. The implementation of synchronization is original, creative, and thought-compelling.
S3. The experiments are diverse, described thoroughly, and seem to make a compelling case for the model's effectiveness.

Weaknesses
W1. It's difficult to tell the how important the different components of this model are to its effectiveness. It's also, frankly, a bit unintuitive to reason about. Is synchronization helpful to the model, or is it implicitly regularizing? More experimental probing of this would be helpful, especially ablation studies.

---

> ### Author Rebuttal · Authors · 2025-07-30
>
> Thank you for your insightful review. Your summary, in particular, was helpful in understanding whether our work was sufficiently well communicated.
>
> ## Re: difficulty in reasoning about the model owing to its moving parts.
> This is a very useful comment, thank you. To this end we ran an ablation study on a simplified maze task in order to elucidate any performance differences when ablating NLMs and synchronization. We ran this ablation study on a 15x15 maze task, using 100K training steps and parameter-matched (+-9M) models that we will add in a new appendix:
>
> | Method | Test Accuracy (%) | Test Solve Rate (%) |
> | :--- | :---: | :---: |
> | CTM | 94.6 $\pm$ 0.7 | 65.9 $\pm$ 5.7 |
> | CTM (No NLMs) | 82.9 $\pm$ 4.4 | 35.0 $\pm$ 7.2 |
> | CTM (No Synch) | 85.1 $\pm$ 0.5 | 37.5 $\pm$ 0.7 |
> | LSTM + Synch | 82.4 $\pm$ 0.9 | 33.8 $\pm$ 3.3 |
>
> Referring specifically to the question of the role of synchronization, it serves three important functions:
> 1.  It **decouples the representation** for downstream use (i.e., attention and projections to predictions) from the raw neuron states ($z^t$), which enables neurons to develop and enhance dynamics without being constrained by the immediate expectations of the representation itself. In other words, the synchronization representation lets a rich representation unfold and stabilize while still allowing the neurons themselves to oscillate and develop patterns over long time scales.
> 2.  It enables **multi-scale temporal sensitivity**, by way of the learnable decay parameters associated with neuron pairs ($r_{ij}$).
> 3.  It enables **higher cardinality representations** since the number of neuron pairs is far greater than the number of neurons. Using pairwise decay parameters unique for each synchronization representation (i.e., there are multiple, and can be many multiple in the future), this further differentiates the functionality of each pairing in time, not just space.
>
> We explored several techniques for extracting usable representations from the latent space of the CTM ($z^t$), including but not limited to: direct linear projections therefrom, MLP-based projections therefrom, gated projections, all of the aforementioned but into some ‘holder’ representation that updated slowly, and even 1D convolutions. None of these offered the same level of simplicity, flexibility, stability, and representational capacity that synchronization representations could. As soon as we started using synchronization we observed an ‘**unlocking**’ to the complexity of the neural dynamics that we had yet to see to such a degree.
>
> ---
>
> ## Re: questions
>
> ### Re: language model.
> We agree with that sentiment. However, the space of language modelling is extremely competitive and not necessarily well-suited as a test-bed for a truly novel modelling approach. The CTM can certainly benefit from the community effort required to transition a new model to a truly competitive one, but we simply do not have the resources to do that on our own. Hence, we focused on interesting experiments that could yield observations and findings that might eventually serve to understand artificial intelligence on a deeper level.
>
> ### Re: why the best tick in the loss function?
> This is an insightful question. You are right in thinking that it is not strictly necessary. The effect of including the best tick is that it promotes certainty in that it drives the best prediction to be even better. We found that this helped bootstrap learning, although it could benefit from further exploration in the future.
>
> ### Re: Emergent phenomena.
> There simply is no additional space in the main text to flesh these out. Further, many of these observations were as a consequence of the “journey” taken to develop this work, and would be difficult to re-explore after having learnt the lessons the observations had on offer. That said, we agree that future experiments into some of these phenomena would be fascinating and are currently investigating related concepts.
>
> ---
>
> We hope that this response addresses your outstanding concerns and answers your questions. It would be extremely helpful to engage with you further, particularly if there is anything you think we could do that would prompt you to consider improving your review score.

---

> > ### Comment · Reviewer_Fj9J · 2025-08-01
> >
> > Thanks for the thoughtful response, and especially the new ablation study. To re-iterate, I think this paper has some clever ideas and evaluates them experimentally with a great deal of insight. I also looked over the novelty concerns raised by another reviewer and am satisfied with how the authors addressed the issue -- the novelty of the paper still stands. I maintain that this is a good paper and will maintain my score (5, accept) at this point.

---

### Official Review · Reviewer_tk25 · 2025-06-22

**Clarity:** 2
**Significance:** 3
**Originality:** 1
**Rating:** 5
**Confidence:** 4

**Summary:**

This paper introduces a deep neural network architecture which uses timing and synchrony between neuronal units to compute. The architecture is end-to-end differentiable, and the authors demonstrate that it can be optimized to perform interesting and computationally challenging tasks, such as 2D mazes and ImageNet 1K classification. The authors also provide interactive examples of their network performing these tasks.

**Questions:**

## Question on Novelty and Originality

My main question concerns the novelty and originality of the work. The authors claim multiple times that they are the first to incorporate neural synchronization within a deep learning context. For example:

- **Lines 46–47**:
  *"This use of neural synchronization yields a fundamentally new type of biologically-motivated latent representation..."*

- **Line 65**:
  *"The CTM introduces neural timing and synchronization as core computational principles."*

However, this appears to overlook previous work that has explicitly explored similar ideas, most notably:

- **[Reichert & Serre (2014)](https://arxiv.org/abs/1312.6115)**
  *Neuronal synchrony in complex-valued deep networks*
  This work directly addresses the use of neural synchrony in deep networks.

- **[Lee et al. (2022)](https://ieeexplore-ieee-org.libproxy.mit.edu/stamp/stamp.jsp?tp=&arnumber=9849162)**
  *Complex-valued neural networks: A comprehensive survey*
  This survey covers numerous architectures that utilize neural timing and synchronization as core computational principles.

Given the existence of this prior work, was the omission of this literature an honest oversight, or was there a specific reason for its exclusion?  If the latter, could the authors clarify how their approach differs from or advances beyond these previous efforts?

The authors also claim, in the abstract, that "Most artificial neural networks simplify neurons by abstracting away dynamics". I am not sure what this means--can the authors clarify? Do they mean abstracting away the complex biophysics dynamics of single-neuron Hodgkin-Huxley models? Because taken at face value, their sentence is certainly false: many hundreds of papers are written every year which use artificial neural networks with dynamical neurons. This has been going on for many years in neuroscience, for example see [Computation Through Neural Population Dynamics – Yas S, Golub MD, Sussillo D, Shenoy KV. *Annual Review of Neuroscience*, 2020.](https://doi.org/10.1146/annurev-neuro-092619-094115)
 review from five years ago.


**To be clear, I believe this could be a strong and valuable paper, provided the inaccurate novelty claims are removed and the related work section is substantially expanded to more accurately contextualize the contribution within existing literature. I would consider increasing my score if the authors were to address my concerns here.**

**Ethical Concerns:**

["NO or VERY MINOR ethics concerns only"]

**Final Justification:**

The authors clearly responded to all my points.

**Limitations:**

Yes.

**Paper Formatting Concerns:**

N/A.

**Quality:**

3

**Strengths And Weaknesses:**

# Strengths:

The paper is well-written, and the technical contributions are clearly described. Also, the authors go above and beyond by providing interactive examples of their network solving a 2D maze task, which this reviewer found charming and fun to play with.

# Weakness:

Unfortunately, I have major concerns about the originality of this work. Although the authors repeatedly claim that they are the first to use neural synchronization in a deep learning context, this is simply untrue (details below in the "Questions" section).

---

> ### Author Rebuttal · Authors · 2025-07-30
>
> Thank you for your insightful and directed review. We appreciate the important points that you have raised regarding novelty and prior work.
>
> ## Re: Clarifying novelty
> We agree that our novelty claim, regarding synchronization, was **insufficiently posited within these prior works**. This lack was an honest oversight, but the specific details of what makes our use of synchronization novel still stand. Specifically, and crucially, we use synchronization as **learnable representations**. These representations are used downstream via weighted projections (e.g., for attention queries and class predictions, although many such representations can be extracted from the singular dynamical system that the CTM defines).
>
> **Reichert & Serre (2014)** introduced a model where synchronization emerges from the interaction of complex-valued neurons, where the mechanism is the phased interaction of complex weighted-sums of inputs, and the role of their synchronization is as a **modulator of information flow**, effectively binding neurons in a form of gating. They make use of these emerging ‘bound’ neuron groups for tasks such as object segmentation, using a post-hoc processing step to use the synchronized neurons. While interesting and useful, this is quite distinct from our direct use of synchronization as a latent representation.
>
> In the context of complex-valued neural networks, there are a variety of different architectures that utilize neural timing and synchronization (Lee et al. 2022). These use “synchronization” in the traditional control and dynamical systems sense: they develop controllers or conditions to make complex-valued neural network models synchronize with each other or reach stable synchronized states. By contrast, the CTM uses neural synchronization in a **computation sense**: it treats the synchronization of neuron activities as a feature representation within a neural network model. There is no external “controller” in CTM enforcing synchronization; instead, CTM’s training optimizes how neurons’ activation patterns synchronize so that this synchrony encodes useful information for tasks. Essentially, CTM repurposes the concept of synchrony as a tool for thinking, whereas the other works treat synchrony as a behavior to be achieved for system stability or control objectives. We believe this makes the CTM fundamentally different from those prior synchronization studies in how the concept is applied.
>
> **Actions:**
> * We will completely clarify that our novel use of synchronization is as a latent, learnable, and internally constructed representation that the model builds and uses downstream. To that end, we propose to rephrase the paragraph starting on line 46 as:
>     > “The CTM learns to use neural synchronization as its latent representation, distinguishing it from existing work that explores synchrony as emergent properties for post-hoc use [citations, including Reichert and Serre]. This representation is distinct from the common static `snapshot' representations used in most modern NNs as it directly encodes the temporal interplay of neural dynamics.”
> * Regarding line 65: this was perhaps a poor phrasing of what we intended to say, as we did not mean to claim that the CTM is the introduction to the use of neural timing, but rather that it uses timing and synchronization as computational tools for its learning. Thank you for pointing that out. We will also rephrase Line 65 to read as:
>     > “The CTM uses neural timing and synchronization as core computational principles...”
> * We propose to include the following paragraph (subject to minor tweaks due to space constraints) in the related work section, dedicated to earlier explorations in synchrony (please feel free to offer critique and/or make requests):
>     > “Reichert & Serre (2014) proposed a model where synchronization emerges from interactions among complex-valued neurons, serving as a gating mechanism that modulates information flow and enables post-hoc grouping of neurons for tasks like object segmentation. Unlike CTM, however, their model does not use synchrony as a learned latent representation during computation. Other approaches in complex-valued neural networks (Lee et al., 2022) employ synchronization from a control-theoretic perspective, aiming to stabilize or coordinate networks via externally enforced synchrony. In contrast, CTM integrates synchronization intrinsically, optimizing neural phase relationships during training to encode task-relevant representations. This positions CTM as a computationally grounded model of synchrony, fundamentally distinct from prior works that treat synchrony as a control objective.”
>
> ---
>
> ## Re: abstracting away dynamics
> Thank you for pointing out this important distinction. We do, indeed, mean single-neuron models, hence the introduction of NLMs, enabling a more complex (learnable) neuron model. We propose to clarify this in the abstract, replacing the sentence you pointed out with the following:
> > “Most artificial neural networks ignore the complexity of individual neurons.”
>
> ---
> ## Additional note: Ablations
> In response to other reviewers we have performed an ablation study to elucidate the impact of NLMs and synchronization. We ran this ablation study on a 15x15 maze task, using 100K training steps and parameter-matched (+-9M) models that we will add in a new appendix:
>
> | Method | Test Accuracy (%) | Test Solve Rate (%) |
> | :--- | :---: | :---: |
> | CTM | 94.6 $\pm$ 0.7 | 65.9 $\pm$ 5.7 |
> | CTM (No NLMs) | 82.9 $\pm$ 4.4 | 35.0 $\pm$ 7.2 |
> | CTM (No Synch) | 85.1 $\pm$ 0.5 | 37.5 $\pm$ 0.7 |
> | LSTM + Synch | 82.4 $\pm$ 0.9 | 33.8 $\pm$ 3.3 |
> ---
> We hope that this sufficiently clarifies our claims and better positions our paper in the full breadth of prior art. Thank you for your excellent suggestions and nuanced questions. Please let us know if we can do anything more, or if you have any further questions.

---

> > ### Comment · Reviewer_tk25 · 2025-08-01
> > **My comments have been addressed.**
> >
> > I appreciate the reviewers for their thoughtful and detailed responses.
> >
> > My primary concern was regarding the paper’s novelty, and the reviewers acknowledge that their contributions could have been more clearly situated within the existing literature. I view this as an honest oversight on the authors’ part, which they have expressed a strong willingness to address.
> >
> > On my end, I was also not exactly clear on what the authors meant by the use of synchronization as a latent representation, but their reply has helped me see the difference between their contribution and prior work.
> >
> > Therefore, with the promised additions and improved contextualization, I am, as previously indicated, raising my score to accept (5).

---

### Official Review · Reviewer_EBTG · 2025-07-01

**Clarity:** 3
**Significance:** 3
**Originality:** 3
**Rating:** 5
**Confidence:** 3

**Summary:**

This paper introduces the Continuous Thought Machine (CTM), a novel artificial neural network model designed to take into account the neural timing and interactions for decision making. The proposed model uses neuron-level processing and synchronization as foundational elements. The proposed method has been tested on several datasets (2D Maze navigation, Imagenet 1K classification, cumulative parity tasks, cifar 10 etc. ) and shows promising results.

**Questions:**

Do the neuron level models used in CTM increase the parameters compared to the conventional ANN’s?

From Figure 6 (Page 8), CTM's training seems sensitive to random seed, initialization. For internal tricks varying from 75 to 100, the variance is comparatively higher.

Have the authors experimented on trying biological neuronal models such as Hindmarsh and rose neuronal model, Hodgkin and Huxley neuronal model etc.? Any specific reason on why biological neuronal models are not used?

The longer training time seems to go against biological plausibility. Would like to know the response of the authors.

**Ethical Concerns:**

["NO or VERY MINOR ethics concerns only"]

**Limitations:**

I would like the authors to respond to the questions raised.

**Paper Formatting Concerns:**

I did not observe any formatting concerns.

**Quality:**

4

**Strengths And Weaknesses:**

Strength

Introducing neural timings and synchronization as the core principle is a novel idea and moves closer to biologically inspired learning algorithms. CTM has been tested on wide datasets and shows promising results. The paper provides extensive details on architecture, hyperparameters, datasets, and optimization settings in the appendices.

Weakness

From Figure 6 (Page 8), CTM's training seems sensitive to random seed, initialization. For internal tricks varying from 75 to 100, the variance is comparatively higher.

The longer training time seems to go against biological plausibility. Would like to know the response of the authors.

---

> ### Author Rebuttal · Authors · 2025-07-30
>
> We sincerely thank you for your thoughtful feedback and are encouraged by the positive assessment of our work. The questions raised are insightful and touch upon some of our core design principles. We hope that addressing these points will strengthen the paper.
>
> ## Weaknesses and key questions
>
> You have highlighted two primary concerns: (1) the CTM's sensitivity to initialization in the parity task and (2) the apparent tension between long training times and biological plausibility. We will address these crucial points first.
>
> ### 1. Training stability/sensitivity to seed
> You are correct that different random seeds can lead to varied final performance. We discuss and analyze this in Appendix F.6.
>
> Crucially, we argue that this variance is not owing to arbitrary instability but rather a direct and fascinating consequence of the CTM discovering **qualitatively different algorithmic strategies** to solve the task. Our analysis in Figure 20 shows that different runs converge to distinct, interpretable algorithms:
> * One successful run learns a **forward-pass strategy**, scanning the sequence from start to end.
> * Another successful run learns a **reverse-pass strategy**, attending to the sequence from end to start.
> * Sub-optimal runs often learn incomplete or less efficient hybrid strategies.
>
> This emergent property actually highlights a **strength of the CTM**: its representational space, built on neural synchronization, is rich enough to enable the discovery of distinct computational solutions. While this can lead to variance in performance depending on the “quality” of the discovered algorithm, it also makes the CTM a powerful tool for investigating *how* models learn to solve procedural tasks, not just *if* they can. The interpretability benefits of the CTM are noteworthy in this case. In the future, we hope to develop techniques for driving towards and/or selecting from the distinct computational solutions the CTM can learn.
>
> To ensure this point is not missed, **we will add a pointer to the main text in Section 6.1** that explicitly references the detailed analysis of these emergent strategies in Appendix F.6.
>
> ---
>
> ### 2. Biological plausibility and training times
> You have made an insightful, and also deeply philosophical point. The comparison between training in deep learning and learning in biology begets a careful consideration of timescales. Consider the following:
> * **Evolutionary and developmental “training”**: The priming of biological brains has occurred over millions of years of evolution, shaping architecture, learning rules, etc. This is followed by years of childhood learning. Taken in totality, such a protracted process is somewhat analogous to the use of millions of gradient descent iterations to train a model’s internal machinery from scratch.
> * **In-situ, rapid learning**: The remarkable few-shot or one-shot learning that humans exhibit (e.g., learning a new object class) is possible only because it builds upon this vast foundation of prior “training”.
>
> Our work aims to capture some functional dynamics of an **“already trained” brain**, rather than simulating its entire evolutionary and developmental trajectory, hence the long training times. Furthermore, a key feature of the CTM is its **native adaptive compute** (Section 5.1). This principle directly aligns with that of biological efficiency at inference time. The CTM naturally learns to “think longer” for challenging inputs, which is to say biologically-plausible ‘rapid’ timescale intelligent resource allocation.
>
> ---
>
> ### 3. Other questions
>
> ### Re: NLM parameter count
> Yes, they do. We acknowledge this trade-off directly in our limitations paragraph (Section 8, line 277).
>
> We view this not as a simple drawback, but as a justified design choice that provides a new axis for model scaling and expressivity. **Per-neuron MLPs** is what enables each neuron to learn a unique, complex internal model, which in turn is fundamental to generating the neural dynamics at the core of the CTM.
>
> ### Re: Neuron model types
> This is an excellent query of our modeling philosophy. Our approach is one of **biological inspiration, not emulation**. We briefly discuss this choice in the conclusion on line 271, and will expand on this as per another reviewer's request.
>
> Models like that of Hodgkin-Huxley are architecturally more realistic representations of the underlying functionality of neurons. These types of neuron models are described by complex, non-linear differential equations. Integrating these models within large-scale, gradient-based deep learning frameworks is, with current methods, computationally prohibitive and often non-differentiable. Taking this approach would have drastically limited the types of experiments we could undertake and would most likely not have yielded the observations we share in this paper.
>
> Instead, we chose to abstract the functional role of a neuron in a dynamic system: to produce a complex temporal output based on its history of inputs. A simple MLP enables the CTM to learn the optimal dynamic response for the task at hand. Crucially, this choice ensures that learning remains tractable, end-to-end differentiable, and compatible with current deep learning paradigms. Our intentional choice in this matter is what allows us to bridge principles (not emulations) from neuroscience with findings and techniques of modern AI.
>
> ---
>
> ## Additional note: Ablations
> In response to other reviewers we have performed an ablation study to elucidate the impact of NLMs and synchronization. We ran this ablation study on a 15x15 maze task, using 100K training steps and parameter-matched (+-9M) models that we will add in a new appendix:
>
> | Method | Test Accuracy (%) | Test Solve Rate (%) |
> | :--- | :---: | :---: |
> | CTM | 94.6 $\pm$ 0.7 | 65.9 $\pm$ 5.7 |
> | CTM (No NLMs) | 82.9 $\pm$ 4.4 | 35.0 $\pm$ 7.2 |
> | CTM (No Synch) | 85.1 $\pm$ 0.5 | 37.5 $\pm$ 0.7 |
> | LSTM + Synch | 82.4 $\pm$ 0.9 | 33.8 $\pm$ 3.3 |
>
> ---
> We hope these responses have clarified our design choices and the novel implications of our work. Please feel free to ask any additional questions or propose changes. Thank you.

---

> > ### Comment · Reviewer_EBTG · 2025-08-03
> >
> > My queries are addressed. I thank the authors for the detailed response.

---

### Official Review · Reviewer_iujh · 2025-07-02

**Clarity:** 2
**Significance:** 4
**Originality:** 4
**Rating:** 5
**Confidence:** 5

**Summary:**

This paper introduces Continuous Thought Machines (CTMs), a novel neural network architecture that emphasizes temporal processing at the single-neuron level and population-level synchronization as core computational principles. CTMs replace standard pointwise nonlinearities with neuron-level models that process a history of pre-activations, aiming to better capture the complexity of biological neural dynamics. Furthermore, CTMs use neural synchronization (defined through correlations in neuron activity across internal “ticks”) as a central representational mechanism for attention and downstream readout. The authors propose that this architecture offers a step toward bridging the gap between biological and artificial intelligence, and present a diverse suite of experiments across different domains, focusing on the qualitative behavior and flexibility of CTMs rather than competitive benchmark performance.

**Questions:**

1. Synchronization as representation: You emphasize synchronization as a key latent representation for attention and output. Could you clarify what precisely is being represented here? Is “synchronization” equivalent to temporal correlation? If not, how does it differ?
2. Decaying influence kernel: In the computation of synchronization, earlier ticks appear to have greater influence due to the exponential decay term $\exp(-r_{ij}(t - \tau))$. This seems counter-intuitive. Can you clarify or justify this design choice?
3. Biological plausibility: Do real neurons retain a history of their pre-activations in a way that could support a mechanism like your neuron-level models? If not, how do you justify this as a biologically inspired component?
4. Subsampling neuron pairs: How sensitive is the model to the random sampling of neuron pairs for computing $\mathbf{S}^t$? How many samples are sufficient, and do you resample at every tick?
5. Ablations: It would be very helpful to see ablation studies isolating the contributions of (a) the neuron-level models and (b) the synchronization mechanism. Could you explore these separately?
6. "Looking Around" Rationale: While an interesting emergent property, what are the specific desired computational benefits or motivations for the CTM to "look around" images during classification, especially for static inputs?

Score could increase if the authors:

- Clarify terminology and figures,
- Provide ablations to better understand the architecture,
- Offer more rigorous comparisons or parameter-controlled baselines, highlighting the trade-offs of CTM vs SOTA models.

**Ethical Concerns:**

["NO or VERY MINOR ethics concerns only"]

**Final Justification:**

This is a technically solid paper and the authors have adequately addressed my remaining questions and concerns in their rebuttal. Therefore, I believe it should be accepted, and I am maintaining my score of 5.

**Limitations:**

Yes.

**Paper Formatting Concerns:**

Figures should be better arranged and better contextualized in the main text, with clearer.

**Quality:**

3

**Strengths And Weaknesses:**

### **Strengths**

- Conceptual originality: The paper proposes a novel architectural shift, where both temporal integration and inter-neuronal synchronization are fundamental to how representations are formed. This is a creative departure from conventional feedforward or RNN-based models.
- Biological inspiration: The authors ground their design choices in neuroscience, particularly in emphasizing the importance of temporal structure and population dynamics, which are often abstracted away in standard deep learning.
- Breadth of experimentation: The authors test CTMs across a range of tasks, demonstrating versatility, even if not aiming for SOTA performance.
- Modularity: The design of neuron-level models, decaying influence kernels, and synchronization mechanisms is well-structured, suggesting future extensibility.

### **Weaknesses**

- Clarity issues:
    - Some key figures (e.g., Figures 1 and 2) are introduced abruptly and not clearly contextualized in the main text. What does “teleporting” mean in Figure 1?
    - Figure 3 is convoluted and could be made clearer.
    - Terms like “neural synchronization as representation" are used repeatedly but never precisely defined, why use the term “synchronization” instead of “temporal correlation”?
- Quantitative rigor: There is a lack of controlled comparisons in the main text with baseline models on shared tasks. It is unclear whether CTMs offer any advantage under equivalent parameter or FLOP budgets.
- Biological grounding: While motivated by neuroscience, some modeling choices (e.g., neurons retaining a full history of pre-activations) lack biological justification. It is unclear whether such memory mechanisms exist at the single-neuron level in biological systems.
- Limited ablation: No ablation experiments are presented to isolate the importance of neuron-level models versus the synchronization mechanism.

---

> ### Author Rebuttal · Authors · 2025-07-30
>
> Thank you for your in-depth review and comments. It is clear that you unpacked and understood both the motivation and efforts of our work.
>
> ---
>
> ## Regarding weaknesses
> ### Clarity issues.
> 1. We have removed the word ‘teleport’. Its use was to capture the process of re-applying the same model sequentially by shifting the start location to the predicted (valid) end location (determined by following the steps predicted by the model). We propose replacing the end of the caption with:
> > ‘... (c) generalizing to 99 x 99 via sequential re-applications of the same model.’
>
> 2. Re Figures 1 and 2.
> **We will contextualize those figures** in the second contribution, providing pointers appropriately. We apologize for this oversight.
>
> 3. Re Figure 3.
> We appreciate this feedback. Unfortunately, visualizing a time-based architecture in a static figure is inherently difficult. This is why we created the supplementary animated video ‘arch.mp4’. In our revision, **we will explicitly reference this video** in the caption, framing it as an essential companion for understanding the model's operational flow.
>
> 4. Re the use of the word “**synchronization**” instead of “**temporal correlation**”.
> We chose “synchronization” because it describes the emergent, coordinated functional behavior we aim to model, a biologically-inspired concept central to population coding in neuroscience. “Temporal correlation”, on the other hand, is the underlying *mathematical mechanism* we use to implement it. To make this point clear, we propose the following change when first introducing the term (in the contributions):
> > “...the use of neural synchronization directly as the representation (implemented via temporal correlations between neuron-level activity; Section 3.4)…”
>
> 5. Re terminology, particularly referring to “**synchronization as a representation**”.
> We explicitly define the meaning of synchronization as a representation using equations in Section 3.4. We now see the need for a slightly more exhaustive descriptive introduction - the above response addresses this.
> ---
> ### Regarding quantitative rigor:
> Thank you for pushing for greater rigor. To that end, **we have taken two direct actions**:
> 1.  We have run ablations (details to follow).
> 2.  We will clarify that our baselines **are already** parameter-matched where applicable.
>
> With that said, we would also like to share the perspective that guided our experimental design. A traditional approach is to immediately position a new idea against SOTA benchmarks. However, we are cognizant that today's leading models are the product of years of compounding engineering gains and fine-tuning on those specific tasks. Given this, we believe that for a fundamentally new architecture like the CTM, a premature focus on SOTA performance would distract from this paper's core scientific contribution: understanding *what* the CTM does and *how* its temporal mechanisms operate.
>
> Our primary goal, therefore, was to establish the *what* and *how* of the CTM's unique operation first, providing a stronger foundation for the community. This required designing novel experiments to specifically probe its capabilities. For example, finding the right “hello world” task was a challenge in itself; our phrasing of the maze task - which requires forming an internal world model by disallowing positional embeddings and outputting a direct action sequence - was a deliberate choice to truly showcase the nuanced differences between the CTM and recurrent baselines like LSTMs. We believe this setup reveals more about our architecture's potential for complex reasoning than a raw performance number on a standard classification benchmark might.
>
> We understand your desire for more rigorous quantitative metrics and hope this response clarifies our rationale and strengthens our contribution.
>
> ---
> ### Regarding biological grounding:
> You have raised an important point about the level of abstraction. Our approach is one of **biological inspiration, not simulation**. The NLM is a functional abstraction of the complex, time-dependent integration that occurs in biological neurons, rather than a literal model of a specific subcellular mechanism. We use a truncated FIFO history (as noted in line 116) as a computationally tractable way to provide this temporal context. We propose the following change to the first paragraph of line 271:
> > “The CTM's NLMs are inspired by the complexity of biological neurons, but are implemented with a level of abstraction appropriate for modern deep learning.”
>
> ---
> ### Regarding limited ablations:
> We have run an ablation study on a 15x15 maze task, using 100K training steps and parameter-matched (+-9M) models that we will add in a new appendix. The results are as follows:
>
> | Method | Test Accuracy (%) | Test Solve Rate (%) |
> | :--- | :---: | :---: |
> | CTM | 94.6 $\pm$ 0.7 | 65.9 $\pm$ 5.7 |
> | CTM (No NLMs) | 82.9 $\pm$ 4.4 | 35.0 $\pm$ 7.2 |
> | CTM (No Synch) | 85.1 $\pm$ 0.5 | 37.5 $\pm$ 0.7 |
> | LSTM + Synch | 82.4 $\pm$ 0.9 | 33.8 $\pm$ 3.3 |
> ---
> ## Regarding questions
> ### Synchronization representation.
> Thank you for asking for this clarification. Section 3.4 defines synchronization mathematically as the temporally-decayed correlation of neuron activity histories. This representation is crucial for three reasons:
> 1.  **Decoupling**: It decouples the representation used for downstream tasks from the raw neuron states ($z_t$), allowing the neurons to develop richer internal dynamics without being solely constrained by the immediate prediction loss.
> 2.  **Hebbian Inspiration**: It captures the idea of “neurons that fire together, wire together.” By making the correlation of activity the representation, we directly model the interplay within the neural population over time.
> 3.  **Multi-scale Temporal Sensitivity**: The learnable decay parameter ($r_{ij}$) for each neuron pair allows the model to learn a representation that is sensitive to phenomena at different timescales simultaneously, which is critical for complex sequential reasoning.
>
> Owing to the lack of space in the main paper, expanding on these points in the main text is not possible. That being said, we can potentially write a new appendix that goes into more detail about the nature of the synchronization representation, should the reviewer request that.
>
> ---
> ### Decaying influence kernel.
> Thank you for this question. The decaying influence kernel gives more weight to recent ticks. As defined in Equation 9, the influence of a past tick, $\tau$, on the current tick, $t$, is scaled by $\exp(-r_{ij}(t-\tau))$. Because the difference $(t-\tau)$ is large for distant ticks and small for recent ones, the negative exponent correctly implements an exponential decay of influence over time.
>
> For example, for $t=3$, the numerator of Equation 10 expands to $z_i(1)z_j(1)\exp(-2r) + z_i(2)z_j(2)\exp(-r) + z_i(3)z_j(3)\exp(0)$ which correctly accounts for a larger influence of the current tick ($z_i(3)z_j(3)$) compared to the previous ticks.
>
> ---
> ### Biological plausibility.
> As discussed in our response to the weaknesses, our goal is **biological inspiration, not direct simulation**. The CTM’s NLMs are an abstraction that enables learning more complexity on a neuron-level, as opposed to an attempt at emulation. Please see the clarification and proposed adjustment to the text (in the weaknesses section, above).
>
> ---
> ### Subsampling neuron pairs.
> * **Sensitivity**: The model is not highly sensitive to the specific random sample of pairs, provided the number of pairs ($D_{out}$, $D_{action}$) is sufficiently large for the task's complexity. For our experiments, we generally chose these values to be on the same order of magnitude as the model's width or the task's output dimensionality.
> * **Resampling**: **The pairs are sampled once at initialization and then fixed.** Resampling at each tick is not feasible, as the projection weights ($W_{out}$, $W_{in}$) are learned specifically for the chosen pairs. **We will clarify this in Section 3.4.1.**
>
> ---
> ### Ablations.
> As mentioned in our response to weaknesses, we have now completed an ablation study that shows NLMs and synchronization are both beneficial.
>
> ---
> ### “Looking around” rationale.
> This is a fantastic question about the functional role of this emergent behavior. When we built the CTM there was never any explicit objective for “looking around”, so when we observed this behaviour we were struck by its parallels to how we expect an intelligent agent to observe a scene. Thus, we believe its emergence points to both a compelling parallel with observable intelligence and a tangible computational benefit.
>
> From a conceptual standpoint, we often identify intelligence functionally through action. A crow solving a puzzle is deemed intelligent because it undertakes actions to achieve a goal. Similarly, the CTM's emergent “gaze” behaviour is an active process of information gathering. That this behavior arises without any direct supervision is a fascinating result because it parallels an intuitive aspect of how intelligent agents interact with their environment.
>
> More crucially, from a computational perspective, this “looking around” process is the mechanism that enables **native adaptive computation**. By iteratively attending to different parts of the image over its internal ticks, the CTM can refine its understanding and adjust its processing time based on the input's complexity. As shown in Figure 5a, this allows it to spend more “thought” (more internal ticks) on difficult images compared to simpler ones. This input-dependent allocation of compute is a direct, beneficial consequence of the architecture's temporal design.
>
> We have added clarification to Section 5.2. as follows:
> > “...It does this entirely without prompting or any guide, implementing computationally beneficial adaptive compute in an intuitive fashion…”
>
> ---
> Thank you again for your thorough review. Please feel free to ask more questions.

---

> > ### Comment · Reviewer_iujh · 2025-08-07
> > **Thanks for your response**
> >
> > The authors have addressed all of my remaining questions and concerns. I am maintaining my score to accept (5).

---

### Note · Authors · 2025-08-12

## An improved paper

We would like to thank the reviewers for their time, insights, and engagements. The actionable items were particularly useful as they directly improved the paper.

In (non-exhaustive) **summary**:
- We ran an ablation study on a small maze task to provide evidence that both NLMs and synchronization as a representation are necessary for the strong performance of the CTM.
- We cleared up clarity concerns regarding some terminology.
- We clarified the level of abstraction we engaged in for "biological plausibility": that of deep-learning amenable design that borrowed conceptually from what we know about neurons in brains.
- Clarified that we are not claiming to be the first to consider synchronisation in a deep learning context, but rather that we are the first to propose *synchronization as a representation* for downstream usage. We also proposed an additional paragraph to include sufficient pointers to literature on the matter.
- Discussed the utility of the two-tick loss function.

We will, of course, adapt the paper to include the necessary terminology updates, extra literature review, and an additional ablation appendix.

Thank you once again for your time and consideration.

---

### Decision · Program_Chairs · 2025-09-17

**Decision:**

Accept (spotlight)

**Comment:**

The paper received recommendations for acceptance across the board. The biggest issue raised during reviewing was that the authors did a poor job in citing prior work. To be clear, I think their response to the reviewer regarding this issue was poor and surprising. This is not the first work to even use learnable synchrony, and I encourage the authors to do a deeper dive into the literature to better understand where the field is on this. Indeed, it seems that the authors didn't follow the citations Reichert & Serre 2014 until now, or else they would have done a better job at this. I recommend the authors fix this for their camera ready version.

With that said, the results and presentation are excellent, and the method is interesting and well-suited for NeurIPS. This is an easy accept.